# Differential hepatic distribution of insulin receptor substrates causes selective insulin resistance in diabetes and obesity

Naoto Kubota[1,2,3,4,*], Tetsuya Kubota[1,3,4,5,*], Eiji Kajiwara[1], Tomokatsu Iwamura[1], Hiroki Kumagai[1], Taku Watanabe[6], Mariko Inoue[1,3], Iseki Takamoto[1,3], Takayoshi Sasako[1], Katsuyoshi Kumagai[7], Motoyuki Kohjima[8], Makoto Nakamuta[8], Masao Moroi[5], Kaoru Sugi[5], Tetsuo Noda[9], Yasuo Terauchi[10], Kohjiro Ueki[1] & Takashi Kadowaki[1]

Hepatic insulin signalling involves insulin receptor substrates (Irs) 1/2, and is normally associated with the inhibition of gluconeogenesis and activation of lipogenesis. In diabetes and obesity, insulin no longer suppresses hepatic gluconeogenesis, while continuing to activate lipogenesis, a state referred to as 'selective insulin resistance'. Here, we show that 'selective insulin resistance' is caused by the differential expression of Irs1 and Irs2 in different zones of the liver. We demonstrate that hepatic Irs2-knockout mice develop 'selective insulin resistance', whereas mice lacking in Irs1, or both Irs1 and Irs2, develop 'total insulin resistance'. In obese diabetic mice, Irs1/2-mediated insulin signalling is impaired in the periportal zone, which is the primary site of gluconeogenesis, but enhanced in the perivenous zone, which is the primary site of lipogenesis. While hyperinsulinaemia reduces Irs2 expression in both the periportal and perivenous zones, Irs1 expression, which is predominantly in the perivenous zone, remains mostly unaffected. These data suggest that 'selective insulin resistance' is induced by the differential distribution, and alterations of hepatic Irs1 and Irs2 expression.

[1] Department of Diabetes and Metabolic Diseases, Graduate School of Medicine, The University of Tokyo, Tokyo 113-8655, Japan. [2] Department of Clinical Nutrition Therapy, The University of Tokyo, Tokyo 113-8655, Japan. [3] Clinical Nutrition Program, National Institute of Health and Nutrition, National Institutes of Biomedical Innovation, Health and Nutrition, Osaka 162-8636, Japan. [4] Laboratory for Metabolic Homeostasis, RIKEN Center for Integrative Medical Sciences, Kanagawa 230-0045, Japan. [5] Division of Cardiovascular Medicine, Toho University, Ohashi Hospital, Tokyo 153-8515, Japan. [6] First Department of Medicine, Hokkaido University School of Medicine, Sapporo, Hokkaido 060-8648, Japan. [7] Animal Research Center, Tokyo Medical University, Tokyo 160-8402, Japan. [8] Department of Gastroenterology, Clinical Research Center, National Hospital Organization Kyushu Medical Center, Fukuoka 810-8563, Japan. [9] Department of Cell Biology, Japanese Foundation for Cancer Research-Cancer Institute, Tokyo 135-8550, Japan. [10] Department of Diabetes and Endocrinology, Yokohama City University, School of Medicine, Kanagawa 236-0004, Japan. * These authors contributed equally to this work. Correspondence and requests for materials should be addressed to N.K. (email: nkubota-tky@umin.ac.jp) or to T.K. (email: kadowaki-3im@h.u-tokyo.ac.jp).

The liver plays a central role in the regulation of glucose homoeostasis; it extracts glucose through the portal vein and synthesizes glycogen and triglycerides (TGs) after food intake, and releases glucose by glycogenolysis or gluconeogenesis during fasting[1]. The liver is known to be the target of insulin, which stimulates the glucose uptake and glycogen accumulation, while inhibiting glycogenolysis and gluconeogenesis.

Insulin receptor substrate (Irs)1 and Irs2 are abundantly expressed in the liver, and interact with downstream molecules such as phosphatidylinositol 3-kinase through their SH2 domains in the metabolic regulation[2–5]. While Irs2 expression is suppressed by insulin at the transcriptional level, Irs1 expression remains intact and is not downregulated by insulin[4,6,7]. Thus, the messenger RNA (mRNA) expression levels of Irs2 increase in the fasting state, when the serum insulin levels are quite low, and immediately decrease after food intake, associated with the increase of the serum insulin levels induced by the food intake. On the other hand, Irs1 levels remain essentially unaltered in both the fasting and fed states. Therefore, it is thought that Irs2 mainly functions under the fasting state and immediately after food intake, whereas Irs1 primarily acts thereafter, that is, in the fed state[8,9].

Hepatic insulin resistance refers to the impaired ability of insulin to suppress hepatic glucose production, despite rather elevated circulating insulin levels[10]. According to the findings of the previous studies, under the condition of total failure of insulin receptor (IR) signalling in the liver, increased gluconeogenesis and decreased lipogenesis would be expected to coexist in the liver[5,8,11,12], representing 'total insulin resistance.' In cases of type 2 diabetes with obesity, however, while the insulin action on the hepatic gluconeogenic pathway is impaired, its action on the hepatic lipogenic pathway remains intact or even becomes exaggerated[13,14], a state referred to as 'selective insulin resistance', in contrast to 'total insulin resistance'. This is a pathogenetic paradox underlying the association of type 2 diabetes with obesity[15].

To address this paradox, we proposed the hypothesis that Irs1-mediated insulin signalling is enhanced, whereas Irs2-mediated insulin signalling is impaired in the liver in type 2 diabetes and obesity. Mice lacking in hepatic Irs2 develop 'selective insulin resistance', whereas mice lacking in hepatic Irs1, or both hepatic Irs1 and Irs2 develop 'total insulin resistance'. Moreover, by investigating the differential insulin signalling in the hepatic periportal (PP) zone, the primary site of gluconeogenesis and hepatic perivenous (PV) zone, the primary site of lipogenesis, we found that insulin signalling was impaired in the PP zone, but rather enhanced in the PV zone, caused by the downregulation of Irs2 in both the PP and PV zones, coupled with intact expression of Irs1, which is predominantly expressed in the PV zone. These data suggest that the differential distribution and alterations of Irs1 and Irs2 cause 'selective insulin resistance'.

## Results

**LIrs1KO and LIrs2KO mice exhibited glucose intolerance.** We first examined the expression levels of Irs1 and Irs2 in liver tissues obtained from living donors for the liver transplantation. In subjects with steatosis, the expression levels of Irs1 were significantly increased, wheras those of Irs2 were significantly decreased (Fig. 1a). Similarly, in the livers of mice on a high-fat (HF) diet, while the Irs1 mRNA expression levels remained unaffected, the mRNA expression levels of Irs2 were markedly downregulated (Fig. 1b). These data suggest the possibility that Irs1-mediated signalling is enhanced, whereas Irs2-mediated signalling is impaired in the livers in the cases of type 2 diabetes

and obesity. To verify this, we fed a HF diet to both LIrs1KO and LIrs2KO mice that we had previously generated[8]. The body weight gains were not significantly different between the control and LIrs1KO mice, and control and LIrs2KO mice under either the normal chow (NC) or HF diet condition (Fig. 1c). In the insulin tolerance test, the glucose-lowering effect of insulin was significantly decreased in both groups of mice (Fig. 1d). The oral glucose tolerance test revealed significantly elevated blood glucose and plasma insulin levels in the LIrs1KO mice (Fig. 1e, left panels). In the LIrs2KO mice, the blood glucose levels were significantly elevated and the plasma insulin levels tended to be elevated (Fig. 1e, right panels). We then carried out the pyruvate tolerance test and found that blood glucose levels were significantly increased in both groups of mice (Fig. 1f). Consistent with these data, the expression levels of PEPCK and G6Pase were significantly elevated in the LIrs1KO mice under both the fasting and fed conditions, and in the LIrs2KO mice, tended to be elevated under the fasting condition (Fig. 1g; Supplementary Fig. 1). These data suggest that both LIrs1KO and LIrs2KO mice exhibit insulin resistance and glucose intolerance under a HF diet.

**LIrs2KO mice but not LIrs1KO mice showed hepatic steatosis.** Macroscopic examination at sacrifice of the animals after 8 weeks of a HF diet revealed a markedly fatty liver in the control and LIrs2KO mice (Fig. 2a, right panels). In contrast, the LIrs1KO mice appeared to be protected from the hepatic steatosis (Fig. 2a, left panels). Oil Red O staining of liver sections also confirmed that the steatosis seen in the control and LIrs2KO mice was prevented in the LIrs1KO mice (Fig. 2b). Consistent with these data, the elevation of the TG content observed in the control and LIrs2KO mice under a HF diet was significantly inhibited in LIrs1KO mice (Fig. 2c,e). Furthermore, *de novo* lipogenesis was significantly decreased in the LIrs1KO, but not LIrs2KO, mice (Fig. 2d,f). We next investigated the expression levels of the genes associated with lipogenesis. The sterol regulatory element-binding protein (SREBP)1c, acetyl-CoA carboxylase (ACC) and fatty acid synthase (FAS) expressions were significantly increased in the fed state in all of the control, LIrs1KO and LIrs2KO mice (Fig. 2g). The degree of increase was similar between the control and LIrs1KO mice, and control and LIrs2KO mice (Fig. 2g). On the other hand, although the expression levels of peroxisome proliferator-activated receptor (PPAR)γ, fat-specific protein (FSP)27 and CD36 did not differ between the fasting and fed condition states in the control, LIrs1KO or LIrs2KO mice, these mRNA expression levels were significantly lower in the LIrs1KO mice (Fig. 2h), which could explain the protection against the development of steatosis and decreased TG content in the liver of LIrs1KO mice (Fig. 2a–d). On the other hand, the expression levels were similar between the control and LIrs2KO mice (Fig. 2h). Although ER stress has been reported to promote the proteolytic cleavage and activation of SREBP1c in the liver independently of insulin signalling[16], the expressions of BIP, CHOP, ATF6 and sXBP were not significantly different between the control and LIrs1KO mice, or control and LIrs2KO mice (Supplementary Fig. 2). These data suggest that LIrs2KO, but not LIrs1KO, mice develop hepatic steatosis under a HF diet.

**The crucial factors in selective insulin resistance.** Although control and LIrs1/2DKO mice showed similar body weights under a NC diet, the body weight gain was significantly lower in the LIrs1/2DKO mice, as compared with that in the control mice under a HF diet (Fig. 3a). LIrs1/2DKO mice on a HF diet showed severe insulin resistance and marked hyperglycaemia after glucose

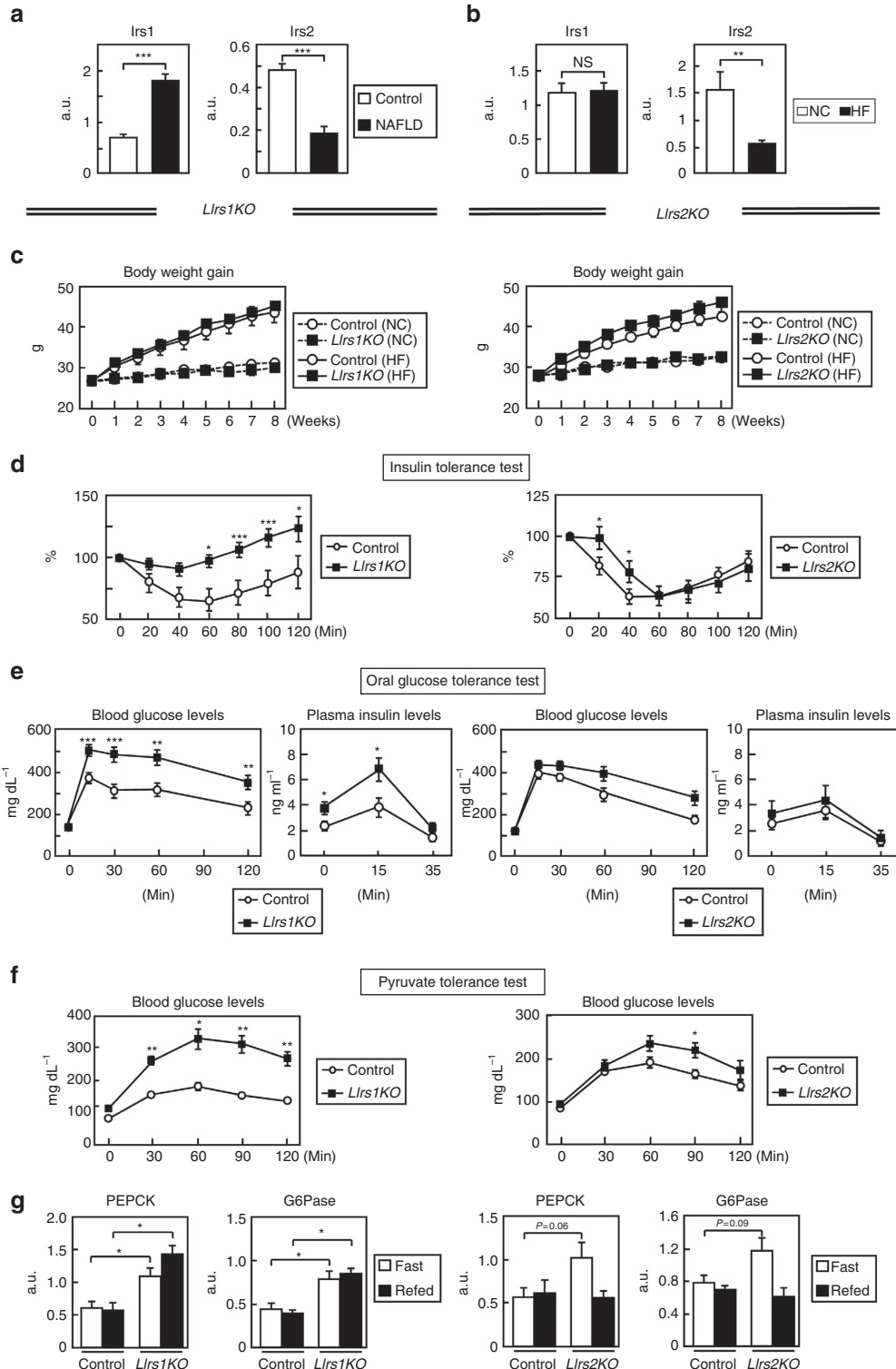

**Figure 1 | LIrs1KO and LIrs2KO mice exhibited insulin resistance and glucose intolerance under a HF diet.** (**a**) Real-time RT–PCR analysis to determine the expression levels of Irs1 and Irs2 in liver biopsy specimens obtained from patients with NAFLD ($n = 30$). (**b**) The expression levels of Irs1 and Irs2 in the livers obtained from mice on NC and HF diet ($n = 12$-$14$). (**c**) Body weight gain in LIrs1KO and LIrs2KO mice on NC or a HF diet ($n = 9$-$20$). (**d**–**f**) Insulin tolerance test (**d**), oral glucose tolerance test (**e**) and pyruvate tolerance test (**f**) in LIrs1KO and LIrs2KO mice on HF diet ($n = 13$-$20$). (**g**) Expression levels of the gluconeogenic genes in the livers of LIrs1KO and LIrs2KO mice in the fasting and fed states ($n = 5$-$6$). Results are represented as mean ± s.e.m. *$P < 0.05$, **$P < 0.01$, ***$P < 0.001$, (Student's $t$-test was used to analyse the statistical significances of differences between two groups, and analysis of variance for the statistical significances of differences among multiple groups).

load (Fig. 3b,c). These mice also showed marked elevation of the blood glucose levels after administration of the gluconeogenic substrate pyruvate (Fig. 3d). Consistent with these data, the

expression levels of PEPCK and G6Pase were significantly elevated in the LIrs1/2DKO mice under both the fasting and fed conditions (Fig. 3e; Supplementary Fig. 3). Hepatic steatosis was

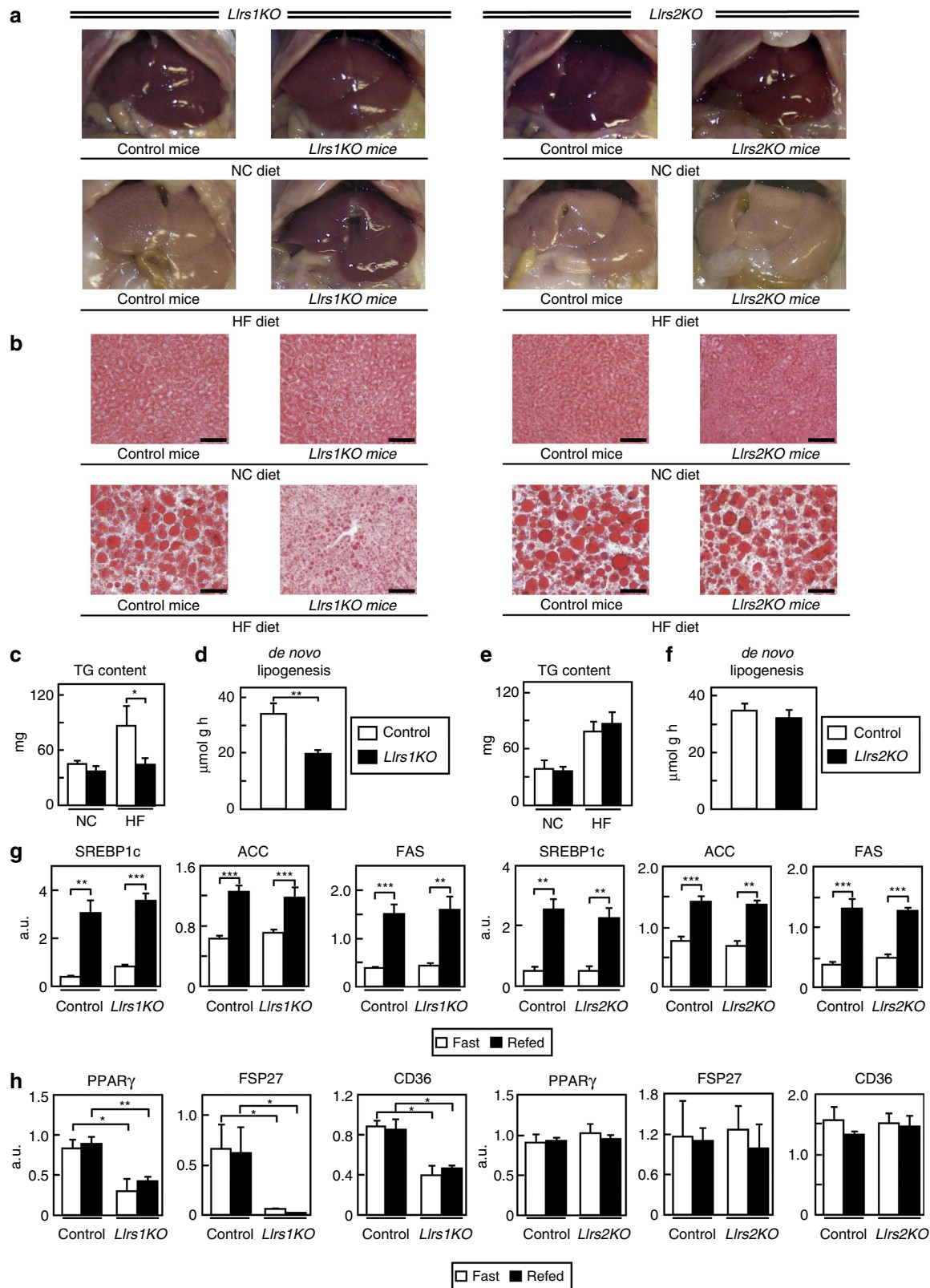

**Figure 2 | *Llrs2KO*, but not *Llrs1KO*, mice showed hepatic steatosis under a HF diet.** (**a**,**b**) Representative gross appearance (**a**) and Oil Red O staining (**b**) (scale bar, 50 μm) of the liver in *Llrs1KO* and *Llrs2KO* mice on NC or a HF diet. (**c**) TG content of the liver in *Llrs1KO* mice on NC or a HF diet ($n = 5$-6). (**d**) *de novo* lipogenesis in the livers of *Llrs1KO* mice on a HF diet ($n = 3$-4). (**e**) TG content of the liver in *Llrs2KO* mice on NC or a HF diet ($n = 5$-6). (**f**) *de novo* lipogenesis in the livers of *Llrs2KO* mice on a HF diet ($n = 3$-4). (**g**) Expression levels of SREBP1c, ACC and FAS genes in the livers of *Llrs1KO* and *Llrs2KO* mice under the fasting and fed states ($n = 5$-6). (**h**) Expression levels of PPARγ, FSP27 and CD36 genes in the livers of *Llrs1KO* and *Llrs2KO* mice in the fasting and fed states ($n = 5$-6). Results are represented as mean ± s.e.m. *$P < 0.05$, **$P < 0.01$, ***$P < 0.001$, (Student's *t*-test was used to analyse the statistical significances of differences between two groups, and analysis of variance for the statistical significances of differences among multiple groups).

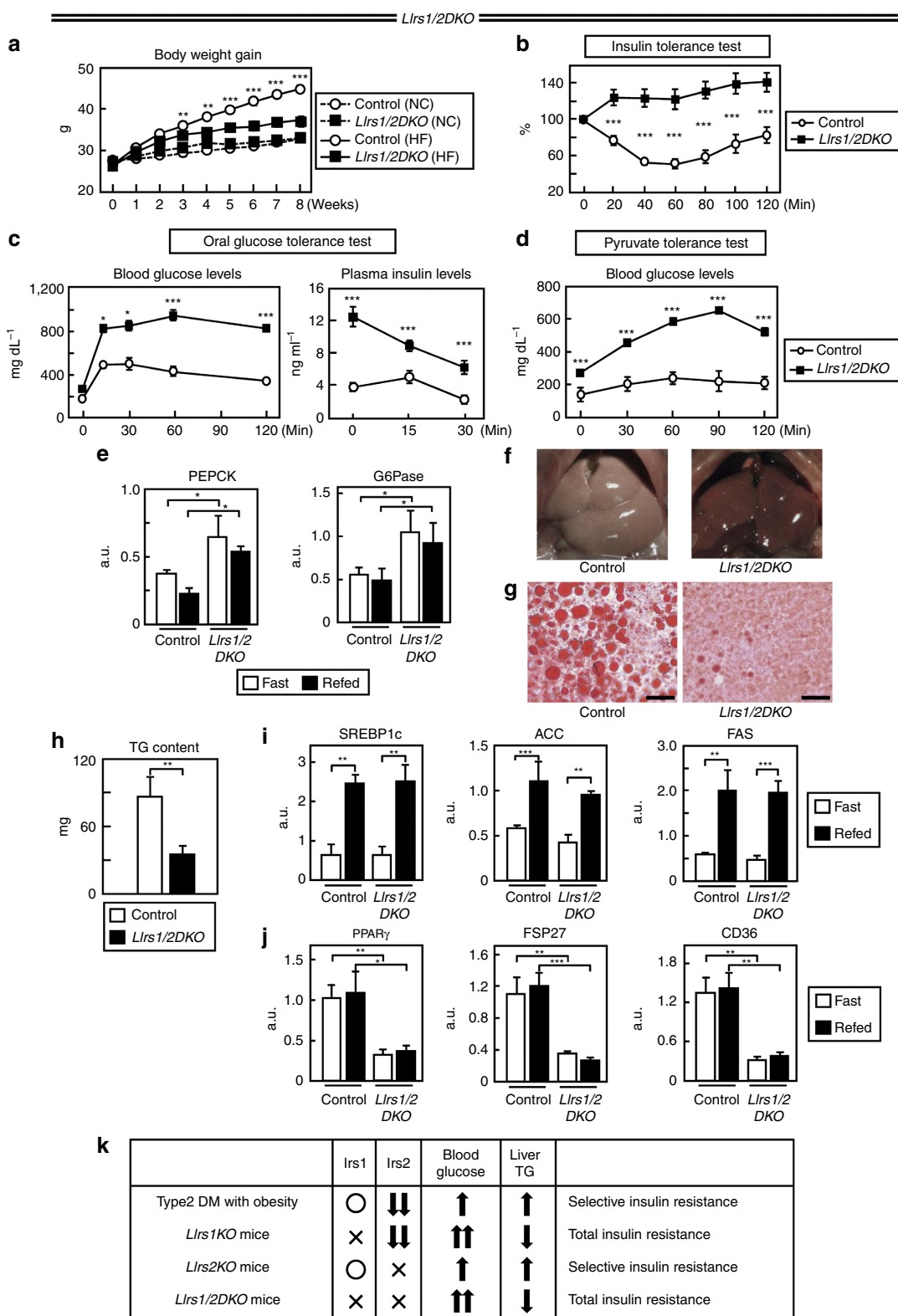

**Figure 3 | Llrs1/2DKO mice showed severe insulin resistance and glucose intolerance, but inhibition of hepatic steatosis.** (**a**) Body weights of Llrs1/2DKO mice on NC or a HF diet (n = 9–10). (**b**–**d**) Insulin tolerance test (**b**), oral glucose tolerance test (**c**) and pyruvate tolerance test (**d**) in the Llrs1/2DKO mice on a HF diet (n = 8–10). (**e**) Expression levels of the gluconeogenic genes in the livers of Llrs1/2DKO mice under the fasting and fed states (n = 5–6). (**f**,**g**) Representative gross appearance (**f**) and Oil Red O staining (**g**) (scale bar, 50 μm) of the liver in the Llrs1/2DKO mice on a HF diet. (**h**) TG content of the liver in Llrs1/2DKO mice on HF diet (n = 6). (**i**) Expression levels of SREBP1c, ACC and FAS genes in the livers of Llrs1/2DKO mice in the fasting and fed states (5–6). (**j**) Expression levels of PPARγ, FSP27 and CD36 genes in the livers of Llrs1/2DKO mice in the fasting and fed states (5–6). (**k**) Comparison of the phenotypes in subject with type 2 diabetes, Llrs1KO, Llrs2KO and Llrs1/2DKO mice. Results are represented as mean ± s.e.m. *P < 0.05, **P < 0.01, ***P < 0.001, (Student's t-test was used to analyse the statistical significances of differences between two groups, and analysis of variance for the statistical significances of differences among multiple groups).

prevented in the *LIrs1/2DKO* mice receiving a HF diet (Fig. 3f,g). In fact, the hepatic TG content was significantly lower in the *LIrs1/2DKO* mice (Fig. 3h). Although the expression levels of SREBP1c, ACC and FAS were similar between the control and *LIrs1/2DKO* mice (Fig. 3i), those of PPARγ, FSP27 and CD36 were significantly decreased in the *LIrs1/2DKO* mice (Fig. 3j). Increased expression levels of SREBP1c, ACC and FAS in the *LIrs1/2DKO* mice to levels similar to those in the control mice in the fed state are probably attributable to the severe hyperglycaemia in the *LIrs1/2DKO* mice, which has been reported to upregulate SREBP1c expression, in addition to insulin signalling[17,18]. To summarize these results (Fig. 3k), because lack of Irs1 in the liver led to 'total' but not 'selective' insulin resistance, despite the downregulated Irs2 expression, Irs1 appears to be essential for the development of 'selective insulin resistance'. In addition, considering the fact that the phenotype associated with the lack of Irs2 in the liver was 'selective insulin resistance', reduced Irs2 expression in the liver may also play a pivotal role in the development of 'selective insulin resistance'. Moreover, mice lacking both Irs1 and Irs2 failed to develop 'selective insulin resistance' despite deletion of Irs2. Taken together, two factors seem to be important in the pathogenesis of 'selective insulin resistance' in the liver: both reduced Irs2 expression and intact Irs1 expression.

**Insulin signalling in the PP and PV zones.** Why did *LIrs1KO* mice show 'total insulin resistance' with hyperglycaemia and suppression of steatosis, whereas *LIrs2KO* mice showed 'selective insulin resistance' with hyperglycaemia and steatosis, despite the high structural homology between the two Irs's and abundant expression of both in the liver? We speculate that the difference in phenotype between the *LIrs1KO* and *LIrs2KO* mice may be attributable to the differential distribution, and alterations of Irs1 and Irs2 expressions in the liver.

A large body of evidence has been accumulated over the years to lend support to the concept of specific zonation for metabolic pathways in the liver, and to suggest that hepatocytes can be divided into two subpopulations, a PP population and PV population, with differing functions; while the PP hepatocytes are involved in gluconeogenesis, which is suppressed by insulin signalling, the PV hepatocytes are involved in glucose uptake and lipogenesis, which is stimulated by insulin signalling[19–24]. We then examined insulin signalling mediated by Irs1 and Irs2 in these two zones of the liver. The efficiency of the hepatocyte separation was confirmed by measuring the mRNA expression levels of marker genes that are well known to show zonal differences in expression, for example, glutamine synthetase, a PV zone marker, and E-cadherin and serine dehydratase, which are PP zone markers[25]. The expression levels of glutamine synthetase were low in the PP zone hepatocytes, but high in the PV zone hepatocytes (Supplementary Fig. 4). In contrast, the expression levels of E-cadherin and serine dehydratase were high in the PP zone hepatocytes, but low in the PV zone hepatocytes (Supplementary Fig. 4). These data confirm the separate identities of the PP and PV hepatocytes in the liver. High expression levels of the gluconeogenic genes, such as PEPCK and G6Pas, were found in the hepatic PP zone of mice receiving a NC diet (Fig. 4a), consistent with the findings of a previous study[25]. High expression levels of the genes for lipid synthesis and storage, such as ACC, PPARγ, Fsp27 and CD36, on the other hand, were found in the hepatic PV zone of mice receiving a NC diet (Fig. 4a). Among the molecules of the insulin signalling pathways, such as the IR, Irs1, Irs2, Akt1 and Akt2, only the expression of Irs1 was twofold higher in the PV zone than in the PP zone (Fig. 4b). Consistent with the results obtained for the mRNA

expression levels, no significant differences in the protein expression levels of IR, Irs2 or Akt protein were found between the PP and PV zones, whereas the Irs1 protein expression level was twofold higher in the PV zone than in the PP zone (Fig. 4c; Supplementary Fig. 5).

We next investigated the mRNA and protein expression levels, and the differences in insulin signalling between the two zones under the HF diet condition. While the IR and, Irs1 mRNA and protein levels remained unaffected in both zones, the mRNA and protein levels of Irs2, which is known to be suppressed at the transcriptional level by hyperinsulinemia associated with HF diet-induced obesity-linked insulin resistance[4,6,7], were markedly downregulated in both the zones (Fig. 4d–f; Supplementary Fig. 5). In the PP zone, although the phosphorylation levels of IR and Irs1 were increased, those of Akt were significantly decreased (Fig. 4e; Supplementary Fig. 5). In the PV zone, in contrast, the phosphorylation levels of IR, Irs1 and also Akt were increased (Fig. 4f; Supplementary Fig. 5). These data indicate that insulin signalling was impaired in the PP zone, but enhanced in the PV zone under the HF diet condition. On the basis of these findings, we hypothesized that under the HF condition where the expression of Irs2 is downregulated, deletion of Irs1 leads to impaired insulin signalling in both the PP and PV zones, resulting in hyperglycaemia and suppression of steatosis. On the other hand, deletion of Irs2 led to the impaired insulin signalling in the PP zone, where Irs1 expression is relatively low and thereby the contribution of Irs2 is relatively greater, leading to the impaired suppression of gluconeogenesis and hyperglycaemia, whereas in the PV zone, where Irs1 expression is relatively high, insulin signalling was maintained despite Irs2 deletion, resulting in steatosis.

**Insulin signalling was impaired in the PP and PV zones in *LIrs1KO* mice.** To test this hypothesis, we next investigated the insulin signalling in the PP and PV zones of the *LIrs1KO* and *LIrs2KO* mice. Irs1 protein expression and phosphorylation of this protein were almost completely abrogated in both the PP and PV zones of the liver in the *LIrs1KO* mice (Fig. 5a; Supplementary Fig. 5). Irs2 protein expression and phosphorylation of this protein were also downregulated in both the PP and PV zones of the liver in the mice under HF diet (Fig. 5a; Supplementary Fig. 5). Consequently, phosphorylation of Akt was significantly decreased in both the hepatic PP and PV zones in the *LIrs1KO* mice (Fig. 5a; Supplementary Fig. 5), suggesting that insulin signalling was impaired in both the PP and PV zones in these mice. Consistent with these findings, the PEPCK and G6Pase expressions were significantly increased in both the hepatic zones in the *LIrs1KO* mice (Fig. 5b). Furthermore, while the expression levels of SREBP1c, ACC and FAS were not significantly different between the control and *LIrs1KO* mice, the expression levels of PPARγ, FSP27 and CD36 were significantly decreased in both the PP and PV zones in the *LIrs1KO* mice (Fig. 5c,d). These data suggest that the lack of Irs1 in addition to downregulation of Irs2 led to the impaired insulin signalling in both the hepatic zones. Thus, knockout of Irs1 in the PP zone impaired the insulin action of suppressing gluconeogenesis, resulting in marked hyperglycaemia, and knockout of Irs1 in the PV zone impaired the insulin action of stimulating lipid synthesis and storage, preventing steatosis (Figs 1d–f and 2a–d).

**Insulin signalling was impaired only in the PP zone in *LIrs2KO* mice.** The phosphorylation levels of IR and Irs1 were slightly, but significantly, increased in both the zones of the *LIrs2KO* mice (Fig. 6a; Supplementary Fig. 6). Irs2 protein expression and its phosphorylation were almost completely abrogated in both the

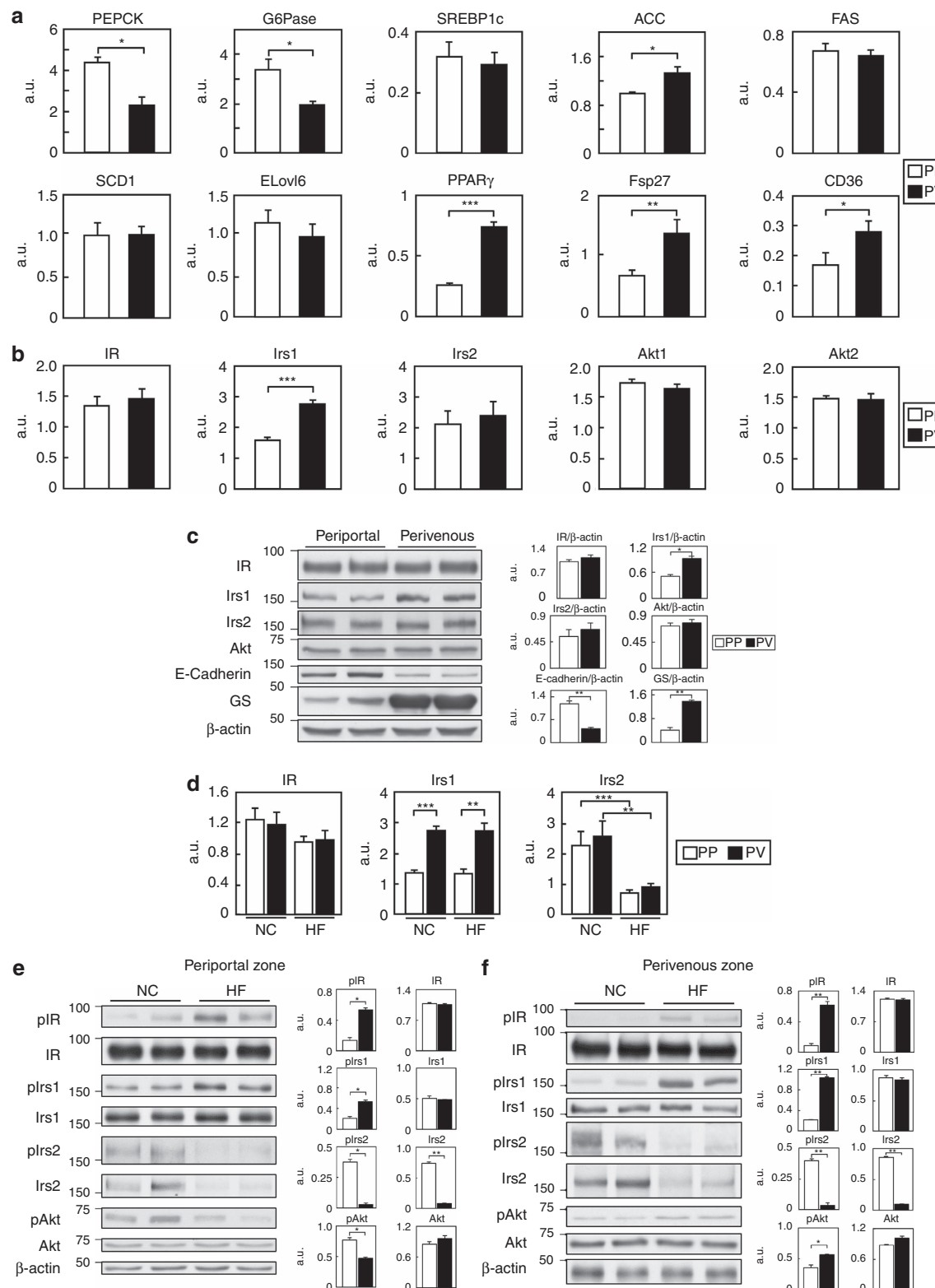

**Figure 4 | Insulin signalling was impaired in the hepatic PP zone, but enhanced in the hepatic PV zone under the HF diet condition.** (**a**) Expression levels of genes for gluconeogenesis and lipid synthesis and storage in the hepatic PP and PV zones of mice on NC diet-fed mice in the fasting state (*n* = 4–6). (**b**) mRNA expression levels of IR, Irs1, Irs2, Akt1 and Akt2 in the hepatic PP and PV zones of NC diet-fed mice under the fasting state (*n* = 4). (**c**) Protein levels of IR Irs1, Irs2 and Akt in the hepatic PP and PV zone of NC diet-fed mice in the fasting state (*n* = 4–5). (**d**) The expression levels of IR, Irs1 and Irs2 mRNA in the hepatic PP and PV zones of NC- or HF diet-fed mice under the fasting state (*n* = 7–9). (**e,f**) Phosphorylation and protein levels of IR Irs1, Irs2 and Akt in the hepatic PP (**e**) and PV (**f**) zones of NC- or HF diet-fed mice under the fasting state (*n* = 6–8). Results are represented as mean ± s.e.m. *$P < 0.05$, **$P < 0.01$, ***$P < 0.001$, (Student's *t*-test was used to analyse the statistical significances of differences between two groups, and analysis of variance for the statistical significances of differences among multiple groups).

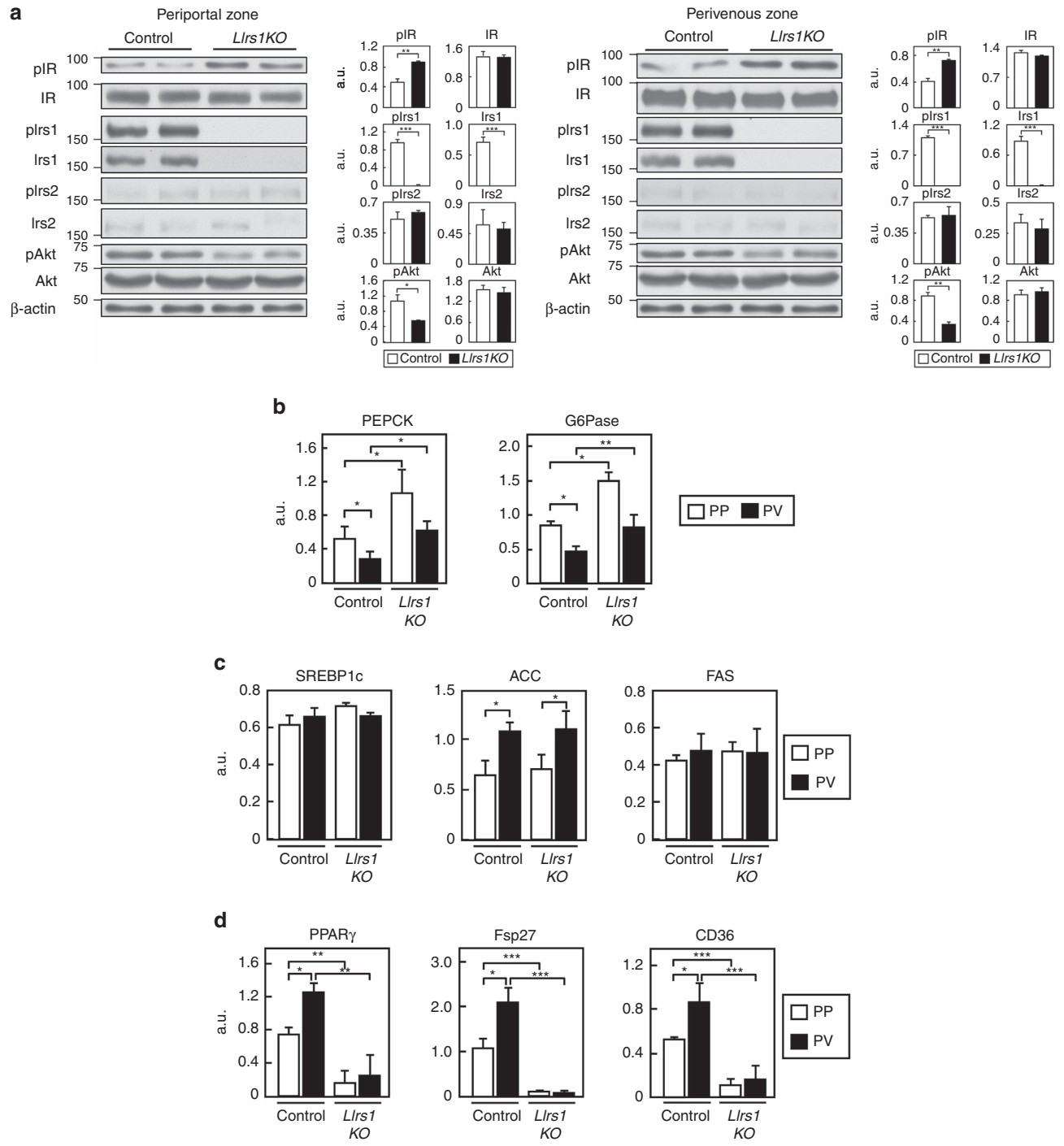

**Figure 5 | Insulin signaling was impaired in the PP and PV zones in the *LIrs1KO* mice under the HF diet condition.** (**a**) Phosphorylation and protein levels of IR Irs1, Irs2 and Akt in the hepatic PP zone of the *LIrs1KO* mice in the fasting state (*n* = 6–8). (**b**) Expression levels of the gluconeogenic genes in the hepatic PP and PV zones of *LIrs1KO* mice in the fasting state (*n* = 5–6). (**c**) Expression levels of SREBP1c, ACC and FAS genes in the hepatic PP and PV zones of *LIrs1KO* mice in the fasting state (*n* = 5–6). (**d**) Expression levels of PPARγ, FSP27 and CD36 genes in the hepatic PP and PV zones of *LIrs1KO* mice in the fasting state (*n* = 5–6). Results are represented as mean ± s.e.m. *$P < 0.05$, **$P < 0.01$, ***$P < 0.001$, (Student's *t*-test was used to analyse the statistical significances of differences between two groups, and analysis of variance for the statistical significances of differences among multiple groups).

PP and PV zones (Fig. 6a; Supplementary Fig. 6). Phosphorylation of Akt was significantly decreased in the PP zone, whereas no significant difference in the Akt phosphorylation level in the PV zone was observed between the control and *LIrs2KO* mice (Fig. 6a; Supplementary Fig. 6). The PEPCK and G6Pase expressions tended to be elevated in the hepatic PP zone in the

*LIrs2KO* mice (Fig. 6b). Although the ACC, PPARγ, FSP27 and CD36 expressions were significantly elevated in the PV zone, their expressions, as well as those of SREBP1c and FAS, were unaltered under the HF diet condition in the control and *LIrs2KO* mice (Fig. 6c,d). These findings suggest that in the PP zone, where Irs1 is less abundantly expressed, lack of Irs2 causes the impaired

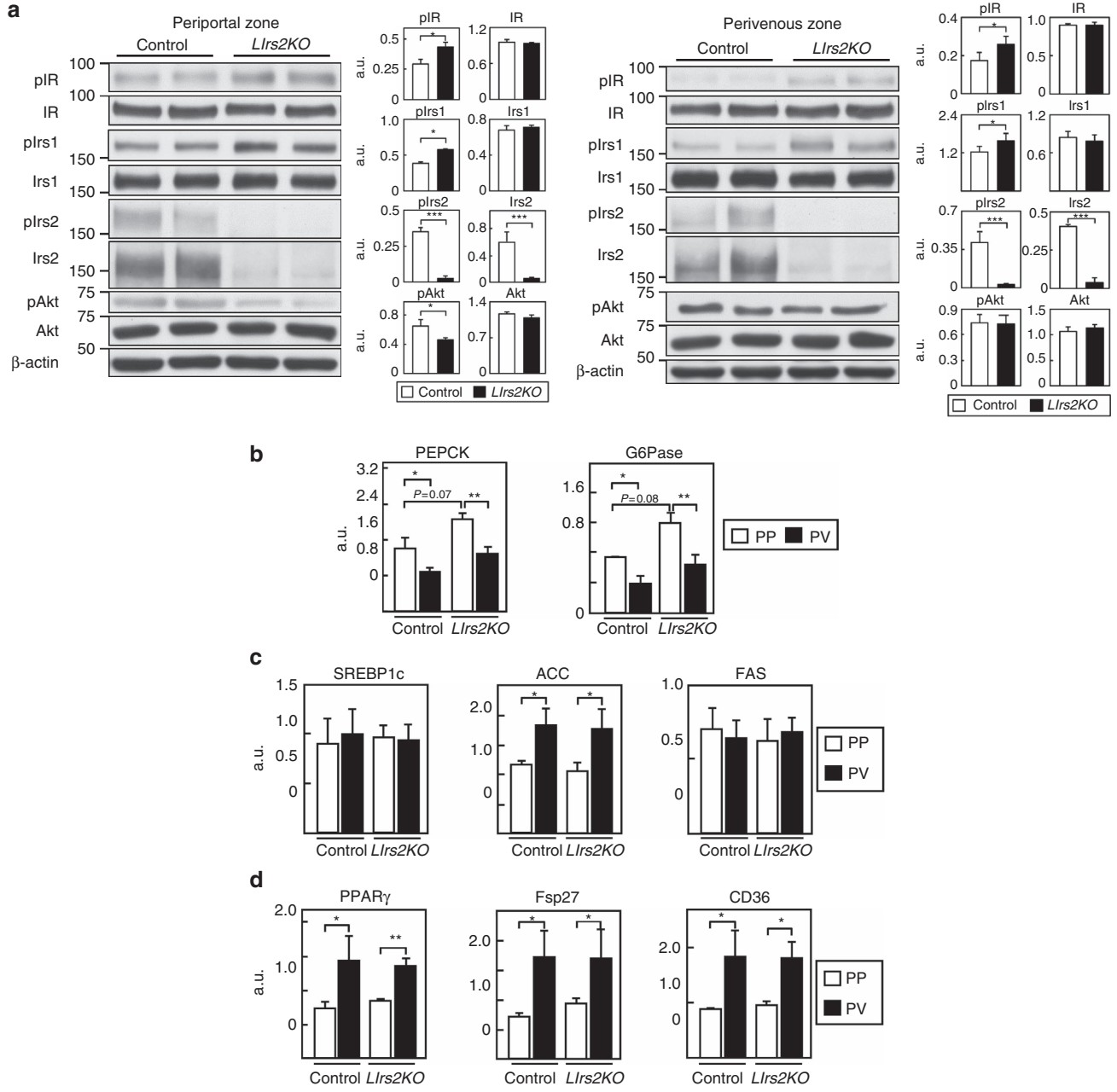

**Figure 6 | Insulin signaling was impaired only in the PP zone in the *LIrs2KO* mice under the HF diet condition.** (**a**) Phosphorylation and protein levels of IR Irs1, Irs2 and Akt in the hepatic PP zone of the *LIrs2KO* mice in the fasting state (*n* = 6–8). (**b**) Expression levels of the gluconeogenic genes in the hepatic PP and PV zones of *LIrs2KO* mice in the fasting state (*n* = 5–6). (**c**) Expression levels of SREBP1c, ACC and FAS genes in the hepatic PP and PV zones of *LIrs2KO* mice in the fasting state (*n* = 5–6). (**d**) Expression levels of PPARγ, FSP27 and CD36 genes in the hepatic PP and PV zones of *LIrs2KO* mice in the fasting state (*n* = 5–6). Results are represented as mean ± s.e.m. *$P < 0.05$, **$P < 0.01$, ***$P < 0.001$, (Student's *t*-test was used to analyse the statistical significances of differences between two groups, and analysis of variance for the statistical significances of differences among multiple groups).

insulin signalling and increased gluconeogenesis, resulting in hyperglycaemia; in contrast, in the PV zone, where Irs1 is abundantly expressed, lack of Irs2 fails to affect insulin signalling, resulting in the maintenance of lipid synthesis, and storage and development of steatosis (Figs 1d–f and 2a,b,e,f).

**Insulin signalling was impaired in the PP and PV zones in *LIrs1/2DKO* mice.** Phosphorylation of IR was significantly increased in both the hepatic PP and PV zones in the *LIrs1/2DKO* mice (Fig. 7a; Supplementary Fig. 6). Irs1 and Irs2 protein

expressions, and their phosphorylation were almost completely abrogated in both the hepatic zones in the *LIrs1/2DKO* mice (Fig. 7a; Supplementary Fig. 6). Consequently, the phosphorylation levels of Akt were also significantly decreased in both the zones (Fig. 7a; Supplementary Fig. 6). The PEPCK and G6Pase expressions were significantly increased in the hepatic PP and PV zones in the *LIrs1/2DKO* mice (Fig. 7b). Although the SREBP1c, ACC and FAS expression levels in the hepatic PP and PV zones were similar between the control and *LIrs1/2DKO* mice, the PPARγ, FSP27 and CD36 expression levels were significantly decreased in both the hepatic zones in the *LIrs1/2DKO* mice

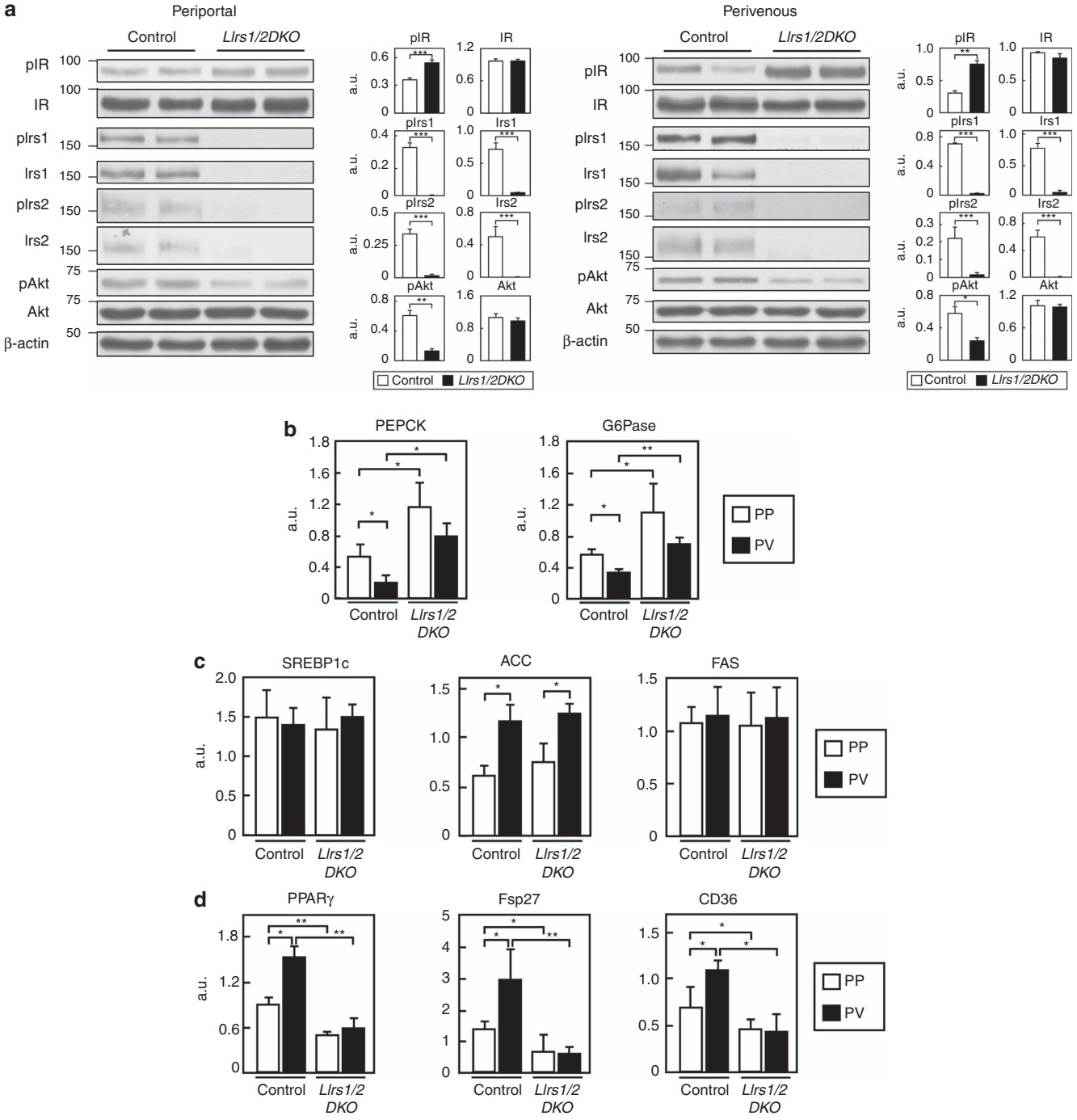

**Figure 7 | Insulin signalling was impaired in both the PP and PV zones of the liver in the *Llrs1/2DKO* mice under the HF diet condition.**
(**a**) Phosphorylation and protein levels of IR, Irs1, Irs2 and Akt in the hepatic PP and PV zones of *Llrs1/2DKO* mice in the fasting state ($n = 6$–8).
(**b**) Expression levels of the gluconeogenic genes in the hepatic PP and PV zones of the *Llrs1/2DKO* mice in the fasting state ($n = 4$–10). (**c**) Expression levels of SREBP1c, ACC and FAS genes in the hepatic PP and PV zones of *Llrs1/2DKO* mice in the fasting state ($n = 4$–10). (**d**) Expression levels of PPARγ, FSP27 and CD36 genes in the hepatic PP and PV zones of *Llrs1/2DKO* mice in the fasting state ($n = 4$–10). Results are represented as mean ± s.e.m. *$P < 0.05$, **$P < 0.01$, ***$P < 0.001$, (Student's *t*-test was used to analyse the statistical significances of differences between two groups, and analysis of variance for the statistical significances of differences among multiple groups).

(Fig. 7c,d). These data suggest that the lack of both Irs1 and Irs2 in the liver significantly interrupts insulin signalling in the PP and PV zones, resulting in 'total insulin resistance', which results in accelerated gluconeogenesis in the PP zone associated with marked hyperglycaemia, and reduced lipid synthesis and storage in the PV zone associated with prevention of steatosis.

**Irs1 expression was regulated by β-catenin/TCF4 signalling.**
The Wnt/β-catenin pathway has been reported to play a key role in liver zonation[24]. While the stabilized and active form of β-catenin is localized in the PV zone hepatocytes, a negative regulator of β-catenin, the adenomatous polyposis coli (APC) protein, is expressed in a complementary manner to inactivate

β-catenin in the PP zone hepatocytes[26,27]. Conditional gene ablation of APC led to expansion of the expression of the PV zone enzymes to all hepatocytes; conversely, when β-catenin signalling was blocked in the liver, the PV zone hepatocytes adopted a PP phenotype[26,27]. In addition, Wnt/β-catenin signalling has been reported to regulate Irs1 transcription[28,29]. Wnt3A or the constitutively active form of β-catenin increased the Irs1 gene and protein expressions, which were conversely decreased by the suppression of the Wnt/β-catenin pathway, including Dickkopf. Chromatin immunoprecipitation (ChIP) analysis also showed that the T-cell factor (TCF)4, which is one of the transcription factors of the lymphoid enhancer factor (LEF)/TCF family that binds to stabilized β-catenin to regulate Wnt signalling[30], is bound to the promoter of Irs1[31]. To verify that Irs1 expression was indeed regulated by β-catenin in the liver, cells from the rat hepatoma cell line H4IIE, which is known to express the active form of β-catenin, were transfected with siβ-catenin, which resulted in an ∼80% reduction of the β-catenin mRNA expression level (Fig. 8a). Transfection of siβ-catenin significantly reduced the mRNA and protein expression levels of Irs1, but not those of Irs2 or TCF4 in these cells (Fig. 8a,b; Supplementary Fig. 6). We next transfected H4IIE cells with siTCF4, which resulted in an ∼90% reduction of the TCF4 mRNA expression level (Fig. 8c). Transfection of siTCF4 significantly reduced the Irs1 mRNA and protein levels, but failed to affect the Irs2, or β-catenin mRNA and protein expression levels in these cells (Fig. 8c,d; Supplementary Fig. 6). These data collectively suggest that the β-catenin/TCF4 pathway regulates Irs1 expression at the transcriptional level. To identify the TCF/LEF response element in the Irs1 promoter, we constructed deletion mutants of the murine Irs1 promoter by progressively deleting portions of the upstream region. The transcriptional activity of each mutant promoter was examined using H4IIE cells (Fig. 8e). The Irs1 promoter at 7,000 bp 5′ from the transcription start site responded to β-catenin/TCF4 in terms of the transcriptional activity, which was completely abrogated by dominant-negative TCF4 transcription of the reporter (Fig. 8e). Deletion of 1,034 bp (between − 7,000 and − 5,966 bp) or more nucleotides from the − 7,000 bp in the Irs1 promoter completely abolished the transcriptional activity of the reporter (Fig. 8e). Thus, the TCF/LEF response elements were confined to the nucleotide region between − 7,000 and − 5,966 in the murine Irs1 promoter. This region contains two adjacent TCF/LEF motifs (Fig. 8f, left panel). To determine whether the β-catenin/TCF4 complex bound to the region containing these motifs, we performed the ChIP assay in the liver using primers flanking the TCF/LEF motif within the Irs1 promoter. We detected a sequence-specific DNA corresponding to the region of the Irs1 promoter in the immunoprecipitates obtained with anti-β-catenin or TCF4 antibody (Fig. 8f, right panel), consistent with the finding reported from a previous study[28]. These data confirm that the β-catenin/TCF4 complex directly binds to the Irs1 promoter and that Irs1 is a target gene of β-catenin/TCF4, contributing to the Irs1 zonal gradient in the liver.

## Discussion

Mice and humans with obesity and type 2 diabetes manifest 'selective insulin resistance' in the liver: insulin fails to suppress gluconeogenesis, while it continues to activate lipogenesis, resulting in the combination of hyperglycaemia and steatosis[15]. An increasing body of evidence supports the notion that hepatic IR signalling is required for the development of steatosis in an insulin-resistant state. Mice with liver-specific IR deficiency exhibited hyperglycaemia and hyperinsulinemia as in other insulin-resistant states, although total ablation of insulin action in the liver protected against the steatosis that would have occurred if the IR signalling was functional[32]. Similarly, although subjects with inactivating mutations in the IR exhibited hyperglycaemia and hyperinsulinemia, they were also protected against the steatosis that usually accompanied insulin resistance when the IR was intact[33]. These studies indicate that in both humans and mice, IR signalling is required in the liver for the development of steatosis in the insulin-resistant state.

Irs1 and Irs2, as also Akt2 are essential for the development of steatosis in the presence of insulin resistance. IR signalling is almost exclusively mediated by Irs1 and Irs2 in the liver[8]. Although LIrs1/2DKO mice showed severe insulin resistance and hyperglycaemia on a HF diet, as observed in obese insulin-resistant mice in general, these mice failed to develop steatosis in this study. Lep^ob/ob mice lacking hepatic Akt2 also failed to amass TGs in their livers despite exhibiting hyperglycaemia[34]. All of these mice described above showed 'total insulin resistance'. In the present study, we demonstrated that the LIrs2KO mice displayed 'selective insulin resistance', which was not observed in either the LIrs1KO or LIrs1/2DKO mice despite these mice also showing downregulation or deletion of Irs2. These data indicate that at least two factors are required for the development of 'selective insulin resistance': reduced Irs2 expression and intact Irs1 expression. Consistent with this notion, Irs1 expression was maintained or rather increased, and Irs2 expression was reduced in the liver biopsy specimens obtained from patients with nonalcoholic fatty liver disease (NAFLD) (Fig. 1a).

Much evidence has been accumulated to support the concept of specific zonation in the liver for metabolic pathways[19–22]. On the basis of their location in relation to the blood vessels, the hepatocytes of each liver lobule can be divided into two subpopulations, an upstream PP population and a downstream PV population[23,24]. Wnt/β-catenin signalling has been reported to play an essential role in the metabolic zonation of the liver[26,35]. A pioneering study carried out by Benhamouche et al.[36] examined inducible liver-specific APC-knockout mice and found that deletion of APC, which is expressed only in the PP compartment of the liver lobule, activated β-catenin in the PP zone to produce gradual loss of zonation. Importantly, blocking β-catenin signalling by ectopic expression of the Wnt inhibitor Dickkopf1 converts PV hepatocytes to the PP phenotype[36]. The data from our study suggest that Irs1 expression, at least in part, is regulated by Wnt/β-catenin signalling, more specifically by the β-catenin/TCF4 complex, associated with higher Irs1 expression in the PV zone than in the PP zone. On the other hand, Irs2 was similarly expressed in the hepatic PP and PV zones, although it was markedly downregulated in both the zones in the animals on a HF diet. Moreover, when we examined the differential insulin signalling in the hepatic PP and PV zones, insulin signalling as reflected by Akt activation, was impaired in the PP zone, but rather enhanced in the PV zone under the HF diet condition. On the basis of these findings, we propose that 'selective insulin resistance' is produced as a result of predominant and intact expression of Irs1 in the PV zone, and downregulation of Irs2 expression in both the zones in the presence of hyperinsulinemia. Expression of Irs1 was higher in the PV zone than in the PP zone, whereas that of Irs2 was downregulated in both the zones. Thus, in the PP zone, where Irs1 is less abundantly expressed and Irs2 expression is downregulated, insulin signalling is impaired despite the hyperinsulinemia, leading to the impaired suppression of gluconeogenesis and hyperglycaemia (Fig. 9). In contrast, in the PV zone, where Irs1 is abundantly expressed, insulin signalling is rather enhanced in the presence of hyperinsulinemia despite the downregulation of Irs2, resulting

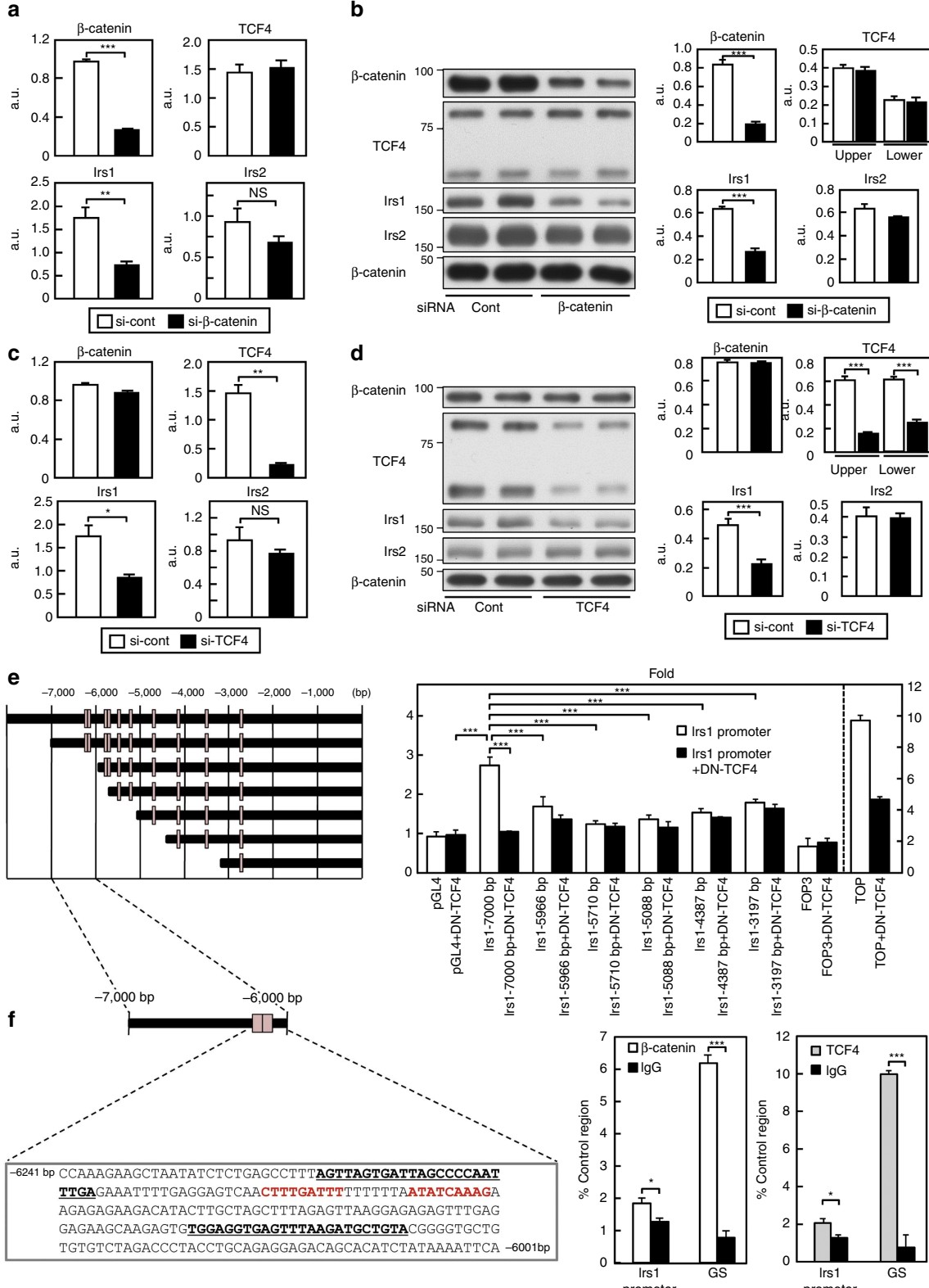

**Figure 8 | Irs1 expression was regulated by β-catenin/TCF4 signalling.** (**a,b**) mRNA (**a**) and protein (**b**) expression levels of β-catenin, TCF4, Irs1 and Irs2 in H4IIE cells transfected with siβ-catenin ($n = 6$–8). (**c,d**) mRNA (**c**) and protein (**d**) expression levels of β-catenin, TCF4, Irs1 and Irs2 in H4IIE cells transfected with siTCF4 ($n = 6$–8). (**e**) Induction of mouse Irs1 promoter activity by the TCF/LEF response element. Results from the representative ($n = 6$) of three replicated experiments is shown. (**f**) Two adjacent TCF/LEF motifs were present within the nucleotide region between $-7{,}000$ and $-5{,}966$ in the murine Irs1 promoter. Endogenous β-catenin or TCF4 chromatin immunoprecipitation with primers against the indicated regions of murine Irs1. GS was used as the positive control region ($n = 4$). The primer sequences are highlighted (bold and underlined). Red text indicates putative TCF4-binding sites. Results are expressed as mean ± s.e.m. *$P < 0.05$, **$P < 0.01$, ***$P < 0.001$ (Student's $t$-test was used to analyse the statistical significances of differences between two groups, and analysis of variance for the statistical significances of differences among multiple groups).

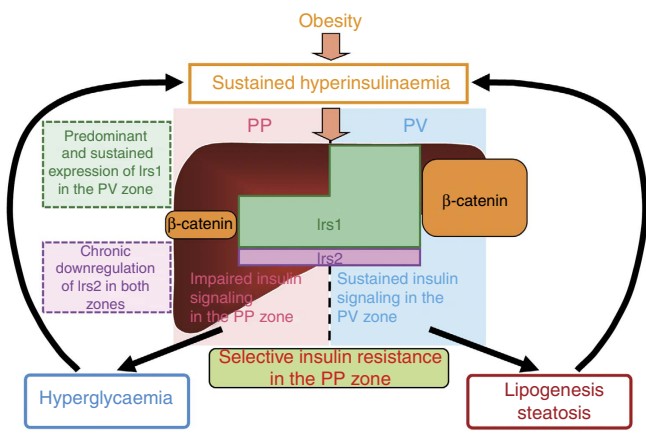

**Figure 9 | Scheme illustrating the mechanism of 'selective insulin resistance' in type 2 diabetes with obesity.** In type 2 diabetes with obesity, while the hepatocyte Irs1 expression levels remain unaffected by the hyperinsulinemia, the expression of Irs2 is downregulated in both the PP and PV zones. Thus, in the PP zone, where Irs1 is less abundantly expressed and Irs2 expression is downregulated, insulin signaling is impaired despite the hyperinsulinemia, leading to the impaired suppression of gluconeogenesis and hyperglycaemia; in contrast, in the PV zone, where Irs1 is abundantly expressed, insulin signaling is rather stimulated in the presence of hyperinsulinemia despite the downregulation of Irs2, resulting in the increased lipogenesis and development of steatosis.

in the increased lipogenesis and development of steatosis (Fig. 9). These data suggest that 'selective insulin resistance' in type 2 diabetes and obesity is caused by the differential distribution, and alterations of hepatic Irs1 and Irs2 expressions.

Sustained insulin signalling in the fasting state appears to be an essential component in the development of 'selective insulin resistance'. When the phosphorylation levels of Akt in the whole liver were measured in the fasting and fed states under the NC and HF diet conditions, the Akt phosphorylation levels were found to be not significantly different in the mice on NC or HF diet in the fed state, whereas in the fasting state, the phosphorylation level was significantly increased in the mice on a HF diet as compared with those on NC diet (Supplementary Figs 6 and 7). These data suggest that while insulin-dependent lipogenesis in the PV zone in the fed state is similar under the NC and HF diet conditions, in the fasting state, it is upregulated under the HF diet condition as compared with the NC diet condition. Elevated insulin signalling in the hepatic PV zone in the fasting state, and not that in the fed state (that is, absence of reduction of insulin signalling even in the fasting state) seems to be responsible for the enhanced lipogenesis under the condition of 'selective insulin resistance'.

Although it has been reported previously that 'selective insulin resistance' is induced by sustained activation of SREBP1c via the mTORC1 pathway even under insulin resistance conditions[18,37,38], we did not find any significant differences in the expression levels of the precursor or mature forms of SREBP1c protein between the control and LIrs2KO mice (Supplementary Figs 6 and 8). Furthermore, the expression levels of SREBP1c were also not significantly different between the hepatic PP and PV zones (Fig. 6c). Thus, 'selective insulin resistance' observed in the LIrs2KO mice cannot be explained by altered SREBP1c expressions or activations. The expression levels of hepatic PPARγ, which regulates the expression levels of many genes controlling fatty acid uptake, fatty acid trafficking, TAG biosynthesis and lipid droplet formation, such as FSP27 and CD36 in the liver[37,39], are usually low in lean mice, but strongly

induced in obese mice[40]. This induction was observed predominantly in the hepatic PV zone, which is the site of lipogenesis. These data suggest that PPARγ located in the hepatic PV zone is likely to play a crucial role in the development of 'selective insulin resistance'. The expressions of PPARγ, FSP27 and CD36 decreased with the suppression of hepatic steatosis in the LIrs1KO mice, whereas they were maintained in the LIrs2KO mice. These data suggest that the aforementioned genes, including PPARγ, are likely to be regulated by insulin signalling, especially by Irs1, in the hepatic PV zone, although simple fasting or refeeding failed to induce any alterations in their expressions (Figs 2h and 3j), unlike the case for SREBP1c (Figs 2g and 3i). Further analysis is needed to clarify how Irs1 and/or Irs2 may be involved in regulating the expression of PPARγ.

What then is the ideal treatment for subjects with 'selective insulin resistance'? Considering the fact that the hyperinsulinemia induced by obesity-linked systemic insulin resistance causes glucose intolerance and hepatic steatosis, simple glycemic control by insulin administration or use of medications to stimulate the insulin secretion is likely to improve the blood glucose levels, but will fail to improve the hepatic steatosis. Amelioration of hyperinsulinemia via reducing the insulin resistance by decreasing the body weight through appropriate modification of the diet and habitual exercise is of utmost importance for subjects with 'selective insulin resistance'. In case of unsatisfactory results obtained by diet and exercise alone, it would be desirable to employ medications to reduce insulin resistance, such as biguanide drugs and thiazolidinediones. Improvement of insulin resistance would be expected to ameliorate hyperinsulinemia, restore Irs2 expression and thereby improve the blood glucose levels. This will further decrease the circulating insulin levels, which would result in reduced lipid synthesis and storage via Irs1 signalling, leading to the improvement of the hepatic steatosis. Thus, treatment is best started from upstream, as shown in Fig. 9.

While considering the mechanism of selective insulin resistance, it is difficult to rule out the possibility that IR signalling diverges downstream of Akt: insulin-induced suppression of gluconeogenesis may be impaired on the one hand, and insulin-induced stimulation of lipogenesis may be activated on the other. In fact, Li et al.[38] reported that there are two insulin signalling pathways in the liver that diverge after Akt and before mTORC1, the latter of which is essential for lipogenesis, but not for the inhibition of gluconeogenesis. It seems likely that both the zonation-dependent mechanism (in this study) and signal divergence-dependent mechanism[38] may contribute to the development of selective insulin resistance.

In this study, we found selective impairment of insulin signalling in the hepatic PP zone in type 2 diabetes and obesity. This may be caused by the downregulation of Irs2 induced by hyperinsulinemia in both the PP and PV zones of the liver, coupled with the higher Irs1 expression levels in the PV zone. Understanding the molecular basis of hepatic insulin signalling and of 'selective insulin resistance' in the dysregulation of glucose and lipid metabolism would also be expected to facilitate a better understanding of the pathogenesis and the treatment of insulin resistance and type 2 diabetes mellitus.

## Methods

**Animals.** Original LIrs1KO, LIrs2KO and LIrs1/2DKO mice (C57BL/6 and 129/Sv mixed background)[8] were backcrossed more than eight times with C57BL/6 mice. All experiments in this study were performed using male littermates at 10–12 weeks of age, and Irs1[lox/lox], Irs2[lox/lox] and Irs1[lox/lox]/Irs2[lox/lox] mice were used as the controls[8,41]. The mice were housed under a 12-h light/dark cycle and given access to food ad libitum. NC (CE-2) and HF (HF-32) diets were purchased from CLEA Japan, Inc. and given to mice for 8 weeks. For the 'fasting condition,' the mice were denied access to food for 16 h, and for the 'fed condition,' the animals were first denied access to food for 16 h, followed by provision of free access to food

for 6 h before the experiments. Genotyping was performed by PCR amplification of the tail DNA from each mouse at 4 weeks of age, as previously reported[8]. The methods used for the animal care and the experimental procedures were approved by the Animal Care Committee of the University of Tokyo.

**Dual-digitonin-pulse perfusion.** PP- or PV-zone-specific hepatocytes were isolated and enriched by digitonin perfusion of the liver, according to the method reported by Taniai et al.[42], with some modifications. The liver was perfused for 10 min with Krebs/Henseleit buffer at 37 °C (pH 7.4). To obtain PP hepatocyte subpopulations, a 3 mg ml$^{-1}$ digitonin solution was infused through the vena cava at a flow rate of 0.3 ml min$^{-1}$ g$^{-1}$ liver. Selective zonal cell destruction was achieved within 30 s of the digitonin perfusion. Washout of digitonin was achieved by perfusion with digitonin-free buffer into the opposite end to the initial digitonin perfusion. To obtain PV hepatocytes, the digitonin solution was infused through the portal vein. The efficiency of hepatocyte separation was confirmed by measuring the mRNA expression levels of marker genes with well-known zonal differences in expression, for example, glutamine synthetase, a PV zone marker, and E-cadherin and serine dehydratase, which are PP zone markers[24]. The expression levels of glutamine synthetase were low in the PP zone hepatocytes, but high in the PV zone hepatocytes (Supplementary Fig. 4). In contrast, the expression levels of E-cadherin and serine dehydratase were high in the PP-enriched hepatocytes, but low in the PV-enriched hepatocytes (Supplementary Fig. 4). These data confirmed successful separation of the PP and PV hepatocytes of the liver.

**Immunoprecipitation and western blot analysis.** To prepare tissue lysates, frozen tissue was homogenized in buffer A (25 mM Tris-HCl, pH 7.4, 10 mM sodium orthovanadate, 10 mM sodium pyrophosphate, 100 mM sodium fluoride, 10 mM EDTA, 10 mM EGTA and 1 mM phenylmethylsulfonyl fluoride). For immunoprecipitation of IR, Irs1 and Irs2, 5 mg portions of the liver extracts were incubated with specific antibodies against IR, Irs1 and Irs2, respectively, for 1 h at 4 °C. Then, protein G Sepharose was added, followed by incubation for 2 h at 4 °C. After washing three times with buffer A, the immunocomplexes were resolved on 7% SDS–polyacrylamide gel electrophoresis. Phosphorylated or total protein was analysed by immunoblotting with specific antibodies against IR (Santa Cruz, Cat. sc-711 1:2,000), Irs1 (Millipore, Cat. 06-248 1:2,000), Irs2 (Millipore, Cat. MAB515 1:2,000) and phosphotyrosine (Millipore, Cat. 05-321 1:2,000). Phosphorylated or total protein expression levels of Akt (pAkt Cell signalling, 9271 1:5,000 and Akt Cell signalling, 9272 1:5,000), E-cadherin (Santa Cruz, sc-4870 1:1,000), glutamine synthetase (Millipore, MAB302 1:5,000) and β-actin (Sigma, A5441 1:5,000) were also analysed by immunoblotting with specific antibodies after the tissue lysates were resolved on SDS–polyacrylamide gel electrophoresis and transferred to a Hybond-P PVDF transfer membrane (Amersham Biosciences, Buckinghamshire, UK). Bound antibodies were detected with horseradish peroxidase-conjugated secondary antibodies using ECL detection reagents (Amersham Biosciences, Buckinghamshire, UK).

***In vivo* glucose homoeostasis.** Glucose tolerance test: mice denied food for 24 h were loaded with oral glucose at 1.5 mg g$^{-1}$ body weight. Blood samples were taken at different time points and the blood concentrations of glucose were measured with an automatic glucometer (Glutest Ace, Sanwa Chemical Co., Nagoya, Japan). Whole blood samples were collected and centrifuged in heparinized tubes, and the separated plasma samples were stored at − 20 °C. Insulin levels were determined using a mouse insulin ELISA kit (Morinaga). Insulin tolerance test: mice were intraperitoneally challenged with 0.75 mU g$^{-1}$ (body weight) of human insulin (Novolin R, Novo Nordisk, Denmark). Venous blood samples were then drawn at different time points[43]. Pyruvate tolerance test: mice denied access to food for 16 h were injected intraperitoneally with pyruvate dissolved in saline (2 g kg$^{-1}$). Venous blood samples were then drawn at different time points.

**Oil Red O staining and TG content of the liver.** Lipid accumulation was assessed by Oil Red O staining of 18-μm frozen sections of the liver fixed in phosphate-buffered 4% paraformaldehyde, according to a previously described method[41] with slight modification. In brief, the livers were washed once for 1 min with $H_2O$. After additional washing for 1 min with 60% isopropanol, the livers were stained for 10 min at 37 °C in freshly diluted Oil Red O solution (six parts of Oil Red O stock solution and four parts of $H_2O$; the Oil Red O stock solution contained 0.5% Oil Red O in isopropanol). For determining the TG content of the liver, tissue homogenate was extracted with 2:1 (vol/vol) chloroform/methanol. Chloroform/methanol was added to the homogenate and the mixture was shaken for 15 min. After centrifugation at 14,000 r.p.m. for 10 min, the organic layer was collected. This extraction was repeated three times, and the collected samples were dried, resuspended in 1% Triton X-100/ethanol and the measurement was conducted using Triglyceride E-test Wako (Wako Pure Chemical Industries Ltd., Osaka, Japan).

**Fatty acid synthesis *in vivo*.** The rate of fatty acid synthesis was measured by a previously reported method[44], with some modification. In brief, each animal was injected intraperitoneally with 50 mCi of [3H] water in 0.1 ml of isotonic saline. One hour after the injection, the animals were anesthetized, and 300–500 ml of blood was removed from the inferior vena cava and used to measure the plasma specific activity of [3H] water in duplicate. The livers were removed, 200–300 mg portions were saponified, and fatty acids were extracted from the samples with 10 ml of petroleum ether after acidification with 1 ml of concentrated HCl[45]; this was followed by a second extraction and evaporation of the petroleum ether. The hepatic fatty acid synthesis rates were calculated as mmol of [3H] water incorporated into fatty acids per hour per gram of tissue[46].

**RNA preparation and RT–PCR.** Total RNA was prepared from the liver using the ISOGEN Reagent Total RNA isolation reagent (Nippon Gene, Tokyo, Japan), in accordance with the manufacturer's instructions. Comparative analysis of the mRNA levels in the liver was performed by reverse transcriptase (RT)–PCR. Total RNA was treated with RNase-free DNase (Nippon Gene, Tokyo, Japan), and first-strand complementary DNA was generated using random 9-mer primers and RT (Takara Shuzo Co., Ltd., Kyoto, Japan). The reverse-transcription mixture was amplified with specific primers. The primers used for IR (Mm 00439693_m1), Irs1 (Mm 00439720_s1), Irs2 (Mm 03038438_m1), PEPCK (Mm 00440636_m1), G6Pase (Mm 00839363_m1), ACC (Mm01304287_m1), FAS (Mm01253300_g1), PPARγ (Mm 00440945_m1), FSP27 (Mm00617672_m1), CD36 (Mm 00432403_m1), glutamine synthetase (Mm00725701_s1), E-cadherin (Mm00486918_m1), serine dehydratase (Mm00455126_m1) and β-actin (Mm00607939_s1) were purchased from Applied Biosystems (CA, USA). The primers for SREBP1c were Fw: 5′-GGAGCCATGGATTGCACATT-3′ and Rv: 5′-GCTTCCAGAGAGGAGGCC-3′. The relative expression levels were compared after normalization to the expression level of β-actin.

**Construction of Irs1 promoter-directed luciferase reporter vectors.** Several DNA fragments containing the mouse Irs1 promoter were PCR amplified from mouse genomic DNA. After verifying their nucleotide sequences by DNA sequencing, the Irs1 promoter fragments were cloned into the luciferase reporter pGL4-Basic vector (Promega, Madison, WI). All the primer sequences are available on request.

**Luciferase assay.** Transfection was carried out at 70–80% confluence of the H4IIE cells (American Type Culture Collection, CRL-1548) using 1.5 μg of Irs1/luciferase reporter gene (pGL3) or TOP Flash or FOP Flash (Addgene 12456 or 12457; Addgene, Cambridge, MA, USA; Randall Moon). The synthetic renilla luciferase reporter vector (phRL-SV40; Promega; 10 ng) was used as the internal control for determining the transfection efficiency. After transient transfection, the cells were collected after overnight serum starvation. Luciferase activity was measured using the Dual-Glo Luciferase Assay System (Promega), according to the manufacturer's protocol.

**ChIP assay.** Lysates of the livers were obtained in buffer containing 1% SDS, 10 mM EDTA, 50 mM Tris-HCl (pH 8.1) and 0.02 tablets per ml of Complete Protease Inhibitor Cocktail (Roche Diagnostics), and subjected to sonication for 20 min at the maximum intensity with 10-s pulses at 4 °C using the Bioraptor (Tosho Denki). They were then diluted 1:10 in buffer containing 1% Triton X-100, 2 mM EDTA, 20 mM Tris-HCl (pH 8.1) and 150 mM NaCl, and precleared with 30 μl of salmon sperm DNA/protein G Sepharose for 2 h at 4 °C. After removal of the Sepharose beads by centrifugation, immunoprecipitation was performed with anti-TCF4 monoclonal antibody (Cell Signaling, Cat. 2565 1:50) or an equal amount of normal rabbit IgG (Cell Signaling, Cat. 2729 1:50), followed by incubation overnight at 4 °C; then, precipitation of the antibody–protein–DNA complexes with salmon sperm DNA/protein G Sepharose was performed for 2 h at 4 °C. The precipitates were sequentially washed with buffers containing 0.1% SDS, 1% Triton X-100, 2 mM EDTA and 20 mM Tris-HCl (pH 8.1) supplemented with either 150 mM (buffer I) or 500 mM NaCl (buffer II), before a final wash in 250 mM LiCl, 1% NP-40, 1% deoxycholate, 1 mM EDTA and 10 mM Tris-HCl (pH 8.1). The pellets were washed with Tris-EDTA buffer and extracted with 1% SDS, 10 mM EDTA and 50 mM Tris-HCl (pH 8.0). After heating at 65 °C overnight, the proteins were digested with proteinase K, and the DNA was purified. The samples were subjected to PCR using the following primers: for Irs1: 5′-AGTTAGTGATTAGCCCCAATTTGA-3′ and 5′-TACAGCATCTTAAACTC ACCTCCA-3′; for a control site located in the Sp5 locus: 5′-GTTGTCTTGGTC ATGGTGTC-3′ and 5′-GACTGTCAAGGAACTAATATAGC-3′.

**Human liver samples.** Tissue samples were obtained by liver biopsy from 30 patients with NAFLD. These patients were admitted to the Kyushu Medical Center or Kyushu University Hospital between 2007 and 2009. As control samples, normal liver tissue specimens were obtained from 30 living donors for liver transplantation, whose liver function tests and histological findings were completely normal. The study protocol was approved by the Ethics Committee of Kyushu Medical Center and Kyushu University Hospital, and written informed consent was obtained from all the patients.

**Nuclear extract preparation.** Livers isolated from mice were homogenized in PBS on ice and centrifuged at 3,000 r.p.m. for 3 min at 4 °C. Pellets were suspended in hypotonic buffer (10 mM HEPES, pH 7.9, 1.5 mM $MgCl_2$, 10 mM KCL, 1 mM DTT and 1× mammalian protease inhibitor cocktail (Sigma)) and disrupted by the addition of 10% NP-40. A small aliquot was taken and saved as the cytoplasmic membrane. The pellets were then resuspended in the nuclear lysis buffer (10 mM HEPES, pH 7.9, 3 mM $MgCl_2$, 100 mM KCL, 1 mM DTT, 0.1 mM EDTA and 1× mammalian protease inhibitor cocktail (Sigma)). The nuclei were pelleted by centrifugation at 15,000 r.p.m. for 15 min at 4 °C, and the purified nuclei were subjected to western blot analysis.

**Statistical analysis.** Values are expressed as the mean ± s.e.m. and were analysed using the JMP software (SAS Institute). Student's $t$-test was used to analyse the statistical significances of differences between two groups, and analysis of variance for the statistical significances of differences among multiple groups. The Tukey–Kramer test was used for post hoc analysis. The statistical significance level was set at $P ≤ 0.05$ in all the tests.

**Data availability.** All data are available from the corresponding author upon request.

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

## Acknowledgements

We thank Tomoko Asano, Eriko Nozaki, Ayumi Nagano, Eishin Hirata, Kousuke Yokota, Yuko Okonogi, Miyoko Suzuki-Nakazawa, Masahiro Nakamaru and Manami Takagi for their excellent technical assistance, and assistance with the animal care. We thank R. Moon for permitting us to use the TOP/FOP flash plasmids. We would also like to express our gratitude to Professor Yoshihiro Maehara and Dr Ken Shirabe of the Department of Surgery and Science, Graduate School of Medical Sciences, Kyushu University, for providing us with the liver samples. This work was supported by a grant for TSBMI from the Ministry of Education, Culture, Sports, Science and Technology of Japan, a Grant-in-Aid for Scientific Research in Priority Areas (A) (16209030), (A) (18209033), and (S) (20229008) from the Ministry of Education, Culture, Sports, Science and Technology of Japan (to T. Kadowaki), a Grant-in-Aid for Scientific Research in Priority Areas (C) (19591037) and (B) (21390279) from the Ministry of Education, Culture, Sports, Science, and Technology of Japan (to N.K.), and the Takeda Science Foundation and ONO Medical Research Foundation (to T.Ku.).

## Author contributions

N.K. and T.Ka. designed this study and wrote the manuscript. N.K., Te.K., E.K., T.I., H.K., T.W., M.I., I.T., T.S. and K.K. conducted the experimental research and analysed the data. M.K. and M.N. contributed to the patient recruitment, and sample collection and analyses. M.N., M.M., K.S., T.N., Y.T. and K.U. contributed to the data discussion. T.Ka. is the guarantor for this work, and as such, had full access to all the data in the study and takes responsibility for the integrity of the data and accuracy of the data analysis. All the authors gave their final approval of the manuscript version submitted for publication.

## Additional information

**Competing financial interests:** The authors declare no competing financial interests.

