## [Peer Review File · Nature Communications]

Redactions:

When text is deleted in rebuttals and referee reports, "[redacted]" has been added in that location.

Reviewers' comments:

Reviewer #1 (Remarks to the Author):

Review of "Selective insulin resistance in type 2 diabetes and obesity is caused by the differential distribution and alterations of insulin receptor substrate (Irs)1 and Irs2 expressions in the liver" by Kubota et al.

In this manuscript, Kubota and colleagues suggest the novel hypothesis that "selective insulin resistance" (i.e. insufficient suppression of glucose production but excessive lipid production) is due to differential effects of Irs1 and Irs2 on periportal metabolic pathways (glucose production) versus perivenous pathways (lipogenesis). This work is original, and of high quality, and is likely to be of great interest to the insulin signaling and metabolism fields.

The authors show that liver knockout of Irs1 (or double knockout of Irs1 and Irs2) causes "total insulin resistance" - as demonstrated by (i) glucose intolerance, insulin intolerance, pyruvate intolerance, and elevated gluconeogenic genes and (ii) decreased lipogenesis, and decreased steatosis. But liver knockout of Irs2 alone causes "selective insulin resistance" - (i) glucose intolerance, insulin intolerance, pyruvate intolerance, and elevated gluconeogenic genes and (ii) increased lipogenesis and steatosis. Based on these findings, the authors propose that "selective insulin resistance" in obese/T2D humans or high fat-fed mice is a setting in which Irs2 signaling is insufficient, but Irs1 signaling is potentiated due to hyperinsulinemia.

The authors propose that the reason this combination of events leads to elevations in both glucose and lipid production is because of spatial organization of these processes. They show that perivenous hepatocytes are enriched for Irs1, Pparg, Fsp27, Cd36, and Acc, and that high fat feeding causes elevated phospho-Akt in this zone. (this is then extrapolated to indicate that these high levels of phospho-Akt lead to high lipogenesis). On the other hand, periportal hepatocytes are enriched for gluconeogenic genes, and high fat feeding decreases Irs2 expression and phospho-Akt in this zone.

Some points that could be addressed to make the conclusions stronger:

-The detailed characterization of the signaling pathways is carried out during the fasting state (length of fast is not stated). Are the alterations in phospho-Akt only present during fasting, or also after feeding?

-Lipogenic gene expression measurements are much more relevant after feeding than fasting. For example, in Figure 4a, they see very little enrichment of Srebp1c, Acc, and Fas in perivenous hepatocytes. But this may be because the samples were from fasted mice and the transcripts were present at only low levels. As another example, in Figure 5a, perivenous zone ("lipogenic zone") of Lirs1KO shows poor phospho-Akt. But there is no defect in Srebp1c, Acc, or Fas, which are downstream of phospho-Akt. Perhaps the reason is because the mice are fasting, thus the levels of these transcripts are low, and thus Akt phosphorylation does not affect them. It is not necessary for all the experiments to be repeated in fed mice, but it is an important factor that has not been addressed.

-The authors use three lipid metabolism-related genes as markers of insulin-stimulated lipid production: Pparg, Fsp27, and Cd36. But these aren't really lipogenic genes; in fact the lipogenic enzymes they measured (Acc and Fasn) don't show much of a preference for Irs1 vs Irs2 knockout or periportal vs perivenous. Could the authors show or describe whether/how Pparg, Fsp27, and Cd36 can explain the effects of selective insulin resistance on lipogenesis or steatosis?

-The authors showed a striking finding but did not comment on it: the induction of Srebp-1c, Acc, and Fas after refeeding is intact in mice lacking Irs1 and Irs2 in liver (figure 3i).

-The length of time fasting and refeeding should be given.

-The insulin and glucose levels should be given for the fasting-refeeding experiments.

- "DN-TCF4" is not defined.

Reviewer #2 (Remarks to the Author):

in the pathogenesis of "selective insulin resistance" in the liver. By investigating differential insulin signaling in the periportal (PP) zone, the site of gluconeogenesis, and perivenous (PV) zone, the site of lipogenesis, of the liver, the authors found that insulin signaling mediated by Irs1 and Irs2 is impaired in the PP zone, but rather enhanced in the PV zone.

These data are interesting as they suggest that "selective insulin resistance" may be induced by the differential distribution and alterations of hepatic Irs1 and Irs2 expressions in type 2 diabetes and obesity.

Major points

- 1-The quality of Oil Red O staining in liver sections should be improved.
- 2- ACC, PPAR γ , Fsp27 and CD36 were found by the authors in the hepatic PV zone of mice. It would be important to determine the zonation of other « lipogenic enzymes » such as SCD1 and Elvol6, whose activity is determinant for hepatic steatosis and insulin signaling.
- 3-Insulin signaling (at the level of IR, Irs1 and Akt) is maintained and/or not decreased in the PV zone of Irs2KO mice and the authors claims that this can explain the « selective » insulin resistance of these mice. However the effects are rather modest. Instead of performing a double KO of Irs1/Irs2, it would have been useful to rescue Irs1 expression in the context of Irs2 KO. Can a selective rescue in the PV zone be considered ?
- 4-Precursor and mature forms of SREBP-1c should be measured, in particular in liver of Irs2 KO mice. Selective insulin resistance was previously reported to occur at the level of sustained activation of SREBP-1c, even in a context of insulin resistance. Mechanisms include ER stress and/or mTORC1 signalling pathways. These components should be measured in livers of Irs2 KO mice.
- 5-Why is ACC expression still induced in livers of LIrs1/2 DKO ?
- 6-In the IrS2 model of selective insulin resistance what is the molecular link explaining sustained lipogenesis? PPAR γ or SREBBP-1c. The authors should discriminate between the two pathways.
- 7- B catenin/TCF4 should be included on the Summary Figure.

Reviewer #3 (Remarks to the Author):

This is a very extensive study comparing the roles of IRS1 and IRS2 in liver insulin resistance. The studies are well done, very exhaustive and support the hypothesis. I have no suggestions for the authors.

Responses to Reviewer #1

We are extremely grateful to Reviewer #1 for the very careful review of our manuscript and for the suggestions that were instrumental in improving the quality of our manuscript.

1. The detailed characterization of the signaling pathways is carried out during the fasting state (length of fast is not stated). Are the alterations in phospho-Akt only present during fasting, or also after feeding?

Lipogenic gene expression measurements are much more relevant after feeding than fasting. For example, in Figure 4a, they see very little enrichment of Srebp1c, Acc, and Fas in perivenous hepatocytes. But this may be because the samples were from fasted mice and the transcripts were present at only low levels. As another example, in Figure 5a, perivenous zone ("lipogenic zone") of Lirs1KO shows poor phospho-Akt. But there is no defect in Srebp1c, Acc, or Fas, which are downstream of phospho-Akt. Perhaps the reason is because the mice are fasting, thus the levels of these transcripts are low, and thus Akt phosphorylation does not affect them. It is not necessary for all the experiments to be repeated in fed mice, but it is an important factor that has not been addressed.

In accordance with the reviewer's suggestion, we examined the phosphorylation levels of Akt in the fasting and fed states under the NC and HF diet conditions. While in the fasting state, the Akt phosphorylation levels were significantly increased in the mice on a HF diet as compared to those on NC diet, in the fed state, no significant difference in the Akt phosphorylation levels was observed in the fed state between the two dietary groups of mice (Supplementary Fig. 5 in the revised manuscript).

[REDACTED]

2. The authors use three lipid metabolism-related genes as markers of insulin-stimulated lipid production: Pparg, Fsp27, and Cd36. But these aren't really lipogenic genes; in fact the lipogenic enzymes they measured (Acc and Fasn) don't show much of a preference for Irs1 vs Irs2 knockout or periportal vs perivenous. Could the authors show or describe whether/how Pparg, Fsp27, and Cd36 can explain the effects of selective insulin resistance on lipogenesis or steatosis?

It has been demonstrated previously that overexpression of PPAR γ in the liver is associated with increased hepatic TG accumulation, regardless of the expression levels of SREBP1c,

ACC and FAS (*Proc. Natl. Acad. Sci. U.S.A.* **109**, 13656-13661, (2012)), which suggests that enhanced expression levels of PPAR γ and its target genes, including FSP27 and CD36, could explain the hepatic steatosis under the condition of “selective insulin resistance,” independent of the expression levels of SREBP1c and its target genes.

3. The authors showed a striking finding but did not comment on it: the induction of Srebp-1c,

Acc, and Fas after refeeding is intact in mice lacking *Irs1* and *Irs2* in liver (figure 3i).

In accordance with the suggestion of the reviewer, we have included our comment on this finding, as follows:

Increased expression levels of SREBP1c, ACC and FAS in the *Lirs1/2DKO* mice to levels similar to those in the control mice in the fed state are probably attributable to the severe hyperglycemia in the *Lirs1/2DKO* mice, which has been reported to upregulate SREBP1c expression in addition to insulin signaling^{17,18} (page 9 lines 157-page 10 lines 160 in the revised manuscript).

4. The length of time fasting and refeeding should be given.

We apologize for not mentioning these in our original text; the fasting duration was 16 hours, and the refeeding duration was 6 hours (page 23 lines 409-411 in the revised manuscript).

5. The insulin and glucose levels should be given for the fasting-refeeding experiments.

In accordance with the reviewer's suggestion, we investigated the insulin and glucose levels in the fasting and fed states in the *Lirs1KO*, *Lirs2KO*, and *Lirs1/2DKO* mice. (Supplementary Figs. 1 and 3 in the revised manuscript)

6. "DN-TCF4" is not defined.

We apologize for not explaining this abbreviation in our original text; "DN" denotes "dominant negative." We have corrected our error in the revised manuscript (page 16 lines 286 in the revised manuscript).

Responses to Reviewer #2

We are extremely grateful to Reviewer #2 for the very careful review of our manuscript and for the suggestions that were instrumental in improving the quality of the manuscript.

1. The quality of Oil Red O staining in liver sections should be improved.

We apologize for the less-than-optimal quality of the Oil Red O staining. In accordance with the reviewer's suggestion, we carried out Oil Red O staining again of liver sections (changed in thickness from 6 to 18 μm) obtained from the *Lirs1KO*, *Lirs2KO*, and *Lirs1/2DKO* mice. (Figs. 2b and 3g in the revised manuscript)

2. ACC, PPAR γ , Fsp27 and CD36 were found by the authors in the hepatic PV zone of mice. It would be important to determine the zonation of other B; lipogenic enzymes such as SCD1 and Elvol6, whose activity is determinant for hepatic steatosis and insulin signaling.

We completely agree with the reviewer's suggestion. In accordance with the reviewer's suggestion, we investigated the expression levels of SCD1 and Elvol6 in the liver and found that the expression levels of these enzymes were not significantly different between the hepatic PP and PV zones (Fig. 4a in the revised manuscript).

3. Insulin signaling (at the level of IR, Irs1 and Akt) is maintained and/or not decreased in the PV zone of *Irs2KO* mice and the authors claims that this can explain the selective insulin resistance of these mice. However the effects are rather modest. Instead of performing a double KO of *Irs1/Irs2*, it would have been useful to rescue *Irs1* expression in the context of *Irs2* KO. Can a selective rescue in the PV zone be considered ?

We appreciate your insightful suggestion. In accordance with the suggestion, we constructed an *Irs1* expression adenovirus vector with the glutamine synthetase promoter to express *Irs1* in the PV zone alone. When the vector was injected first into the *Lirs1KO* mice, *Irs1* was unfortunately expressed not only in the PV zone, but also in the PP zone. To the best of our knowledge, there have been no previous reports in the literature of zone-specific expression of any genes in the liver. Despite our best efforts, we could not achieve *Irs1* expression exclusively in the PV zone, and we apologize for failing to accomplish this.

4. Precursor and mature forms of SREBP-1c should be measured, in particular in liver of *Irs2* KO mice. Selective insulin resistance was previously reported to occur at the level of sustained activation of SREBP-1c, even in a context of insulin resistance. Mechanisms include ER stress and/or mTORC1 signalling pathways. These components should be measured in livers of *Irs2* KO mice.

We completely agree with the reviewer's suggestion. Accordingly, we measured the expression levels of the precursor and mature forms of SREBP1c protein in the *Lirs2KO* mice in both the fasting and fed states. The expression levels were similar between the control and *Lirs2KO* mice (Supplementary Fig. 6 in the revised manuscript). In addition, in accordance with the reviewer's suggestion, we investigated the severity of ER stress in the livers of the *Lirs2KO* mice. Our results revealed that the expression levels of BIP, CHOP, ATF6 and sXBP were similar between the control and *Lirs2KO* mice (Supplementary Fig. 2 in the revised manuscript).

5. Why is ACC expression still induced in livers of *Lirs1/2* DKO ?

High glucose levels have been reported to stimulate lipogenesis, independent of SREBP1c expression and insulin signaling activity (*Nat. Rev. Mol. Cell Biol.***16**, 678-689 (2015)). In accordance with the suggestion of the reviewer, we have added the following sentence in the revised manuscript:

Increased expression levels of SREBP1c, ACC and FAS in the *Lirs1/2DKO* mice to levels similar to those in the control mice in the fed state are probably attributable to the severe hyperglycemia in the *Lirs1/2DKO* mice, which has been reported to upregulate SREBP1c expression in addition to insulin signaling^{17,18} (page 9 lines 157-page 10 lines 160 in the revised manuscript).

6. In the *IrS2* model of selective insulin resistance what is the molecular link explaining sustained lipogenesis? PPAR γ or SREBBP-1c. The authors should discriminate between the two pathways.

We greatly appreciate your insightful suggestion. It is an important issue to be addressed, as the precise molecular mechanisms still remain unclear. We have added the following paragraph in the Discussion section based on the data mentioned above (page 20 lines 358-375 in the revised manuscript).

----- Although it has been reported previously that "selective insulin resistance" is induced by

sustained activation of SREBP1c via the mTORC1 pathway even under insulin resistance conditions^{17,38,39}, we did not find any significant differences in the expression levels of the precursor or mature forms of SREBP1c protein between the control and *Lirs2KO* mice (Supplementary Fig. 6). Furthermore, the expression levels of SREBP1c were also not significantly different between the hepatic PP and PV zones (Fig. 5g). Thus, “selective insulin resistance” observed in the *Lirs2KO* mice cannot be explained by altered SREBP1c expressions or activations. The expression levels of hepatic PPAR γ , which regulates the expression levels of many genes controlling fatty acid uptake, fatty acid trafficking, TAG biosynthesis and lipid droplet formation, such as FSP27 and CD36 in the liver^{38,40}, are usually low in lean mice, but strongly induced in obese mice⁴¹. This induction was observed predominantly in the hepatic PV zone, which is the site of lipogenesis (Fig. 5h). These data suggest that PPAR γ located in the hepatic PV zone is likely to play a crucial role in the development of “selective insulin resistance.” The expressions of PPAR γ , FSP27 and CD36 decreased with suppression of hepatic steatosis in the *Lirs1KO* mice, whereas they were maintained in the *Lirs2KO* mice. These data suggest that the aforementioned genes, including PPAR γ , are likely to be regulated by insulin signaling, especially by Irs1, in the hepatic PV zone, although simple fasting or refeeding failed to induce any alterations in their expressions, unlike the case for SREBP1c. Further analysis is needed to clarify how Irs1 and/or Irs2 may be involved in regulating the expression of PPAR γ . -----

7. B catenin/TCF4 should be included on the Summary Figure.

In accordance with the reviewer’s suggestion, we have included β -catenin in the summary figure (Figure 8) in the revised manuscript.

REVIEWERS' COMMENTS:

Reviewer #1 (Remarks to the Author):

Review of NCOMMS-16-02761 by Kubota et al.

This manuscript presents the interesting possibility that selective insulin resistance during obesity is caused by zonation-specific deficits of *Irs2* signaling (in the periportal "gluconeogenic zone") with simultaneously enhanced *Irs1* signaling, downstream of hyperinsulinemia (in the perivenous "lipogenic zone"). This work supports a fresh model of selective insulin resistance.

The revised manuscript did substantially address the criticisms of the first round of review. I only have one remaining comment. Throughout the manuscript, the authors use the terminology of "lipogenesis" to refer to the mechanism by which IRS1 in the PV zone enhances steatosis. However, they clearly show that the lipogenesis enzymes that they investigated (*Acc*, *Fas*, *Scd1*, *Elovl6*, and their regulator *Srebp-1c*) cannot explain the phenotypes. Rather, the authors show that their phenotypes are correlated with changes in *Pparg* and two of its targets: *Fsp27* (a lipid droplet protein) and *Cd36* (a lipid transport protein). As they state in the discussion, *Pparg* "regulates the expression levels of many genes controlling fatty acid uptake, fatty acid trafficking, TAG biosynthesis, and lipid droplet formation." As the authors cannot directly link their phenotypes to the process of lipogenesis, I suggest they choose different terminology to explain the effects of IRS zonation on steatosis.

One minor comment: on Page 14, line 251, the authors refer to Fig. 4a, and I believe they intended to refer to Fig. 6a.

Reviewer #2 (Remarks to the Author):

Although the authors did not fully address the comments raised during the first round of revision, I am mostly satisfied by their answers. I have no further questions/comments.

Responses to Reviewer #1

We are extremely grateful to Reviewer #1 for the very careful review of our manuscript and for the suggestions that were instrumental in improving the quality of the manuscript.

1. The revised manuscript did substantially address the criticisms of the first round of review. I only have one remaining comment. Throughout the manuscript, the authors use the terminology of "lipogenesis" to refer to the mechanism by which IRS1 in the PV zone enhances steatosis. However, they clearly show that the lipogenesis enzymes that they investigated (Acc, Fas, Scd1, Elovl6, and their regulator Srebp-1c) cannot explain the phenotypes. Rather, the authors show that their phenotypes are correlated with changes in Pparg and two of its targets: Fsp27 (a lipid droplet protein) and Cd36 (a lipid transport protein). As they state in the discussion, Pparg "regulates the expression levels of many genes controlling fatty acid uptake, fatty acid trafficking, TAG biosynthesis, and lipid droplet formation." As the authors cannot directly link their phenotypes to the process of lipogenesis, I suggest they choose different terminology to explain the effects of IRS zonation on steatosis.

We completely agree with the reviewer's suggestion. In accordance with the reviewer's suggestion, we replaced the term "lipogenesis" with "lipid synthesis and storage" (lines 171-172, line 209, line 223, line 236, lines 364-365, and line 763 in the second revision of our manuscript)

2. One minor comment: on Page 14, line 251, the authors refer to Fig. 4a, and I believe they intended to refer to Fig. 6a.

We apologize for the error, and have now corrected it (line 229 in the second revision of our manuscript).

Responses to Reviewer #2

We are extremely grateful to Reviewer #2 for the very careful review of our manuscript and for the suggestions that were instrumental in improving the quality of the manuscript.

1. Although the authors did not fully address the comments raised during the first round of revision, I am mostly satisfied by their answers. I have no further questions/comments.

We thank Reviewer #2 for your interest and comment.